# The Daily Mile: Whole-school recommendations for implementation and sustainability. A mixed-methods study

Emily Marchant[1]*, Charlotte Todd[1], Gareth Stratton[2], Sinead Brophy[1]

**1** Medical School, Swansea University, Swansea, United Kingdom, **2** Applied Sport Technology Exercise and Medicine Research Centre, College of Engineering, Swansea University, Swansea, United Kingdom

\* E.K.Marchant@swansea.ac.uk

**Data Availability Statement:** The data from this research study are not publicly available due to concerns of participant confidentiality. Data from this research study contain information that are

## Abstract

Regular physical activity (PA) during childhood is associated with a range of positive health outcomes and higher educational attainment. However, only 2.0% to 14.7% of girls and 9.5% to 34.1% of boys are meeting the current PA guidelines of 60 minutes of moderate to vigorous PA daily. Schools are targeted as a key setting to improve children's PA levels. The Daily Mile (TDM), a teacher-led 15 minute PA intervention was established in 2012 and has been widely adopted globally. However, the dynamic school environment generates challenges for school-based interventions to follow a uniform implementation method resulting in sustainability issues and limited evaluation. The aims of this mixed-methods study were to (1) explore whether whole-school experiences of TDM were related to implementation and (2) examine the association between TDM and CRF in children from high and low socio-economic groups. Focus groups with pupils (n = 6) and interviews with teachers (n = 9) and headteachers (n = 2) were conducted to explore factors associated with successful implementation. Pupils (n = 258 imputed) aged 9–11 from six primary schools in south Wales, United Kingdom participated in CRF assessments (20m shuttle run test) at two time-points (baseline, 6 month follow-up). Thematic analyses of qualitative measures and linear regression analyses of quantitative measures were used to assess the research questions. Qualitative findings identified implementation factors associated with a positive experience of TDM; flexible and adaptable, not replacing current play provision but delivered as an additional playtime, incorporate personal goal setting, teacher participation, whole-school delivery with community support. Both groups demonstrated equal increases in shuttles between baseline and follow-up (deprived: 4.7 ± 13.4, non-deprived: 4.8 ± 16.0). There was no significant difference in this increase for deprived compared to non-deprived children adjusted for age and gender. Findings from this study provide a set of recommendations for the future implementation and sustainability of TDM.

identifiable at both the school and individual level. Ethical approval for this research study was granted by the College Medicine Research Ethics Committee, Swansea University (approval number 2017-0009A, email sumsresc@swansea.ac.uk), on the basis that participants' data was only accessible by the research team. The participants did not consent to having their data publicly available. Requests for access to the data may be directed to the first author by emailing E.K. Marchant@swansea.ac.uk and citing 'Daily Mile 2017-0009A dataset'. Accession codes will be provided upon request for the data.

**Funding:** This work was supported by the Economic and Social Research Council [grant number ES/J500197/1 to EM] https://esrc.ukri.org/ and National Centre for Population Health and Wellbeing Research. The funders had no role in study design, data collection and analysis, decision to publish, or preparation of the manuscript.

**Competing interests:** The authors have declared that no competing interests exist.

# Introduction

Establishing healthy behaviours such as regular physical activity (PA) in childhood is important for maintaining healthy habits through to adulthood. During childhood, regular PA is associated with reduced body fat, more favourable cardiovascular and metabolic disease risk profiles, enhanced bone health and reduced symptoms of anxiety and depression[1]. Activity status during childhood is predictive of PA levels during adulthood[2] and benefits of regular PA during adulthood include a reduced risk of heart disease, stroke, diabetes, breast cancer, colon cancer and 20–30% lower risk of all-cause mortality[1]. The benefits of regular PA are not limited to health outcomes. Research has demonstrated the association between higher levels of moderate to vigorous physical activity (MVPA) and increased educational attainment [3]. PA is also the principle, modifiable determinant of cardiorespiratory fitness (CRF)[4], which reflects the cardiovascular and respiratory system's capacity to supply oxygen during long-term PA[5]. Higher levels of CRF during childhood have been associated with a range of positive health outcomes similar to those of regular PA such as cardiovascular health. Research has demonstrated the relationship between PA and CRF in children regardless of gender, age, ethnicity, economic status and school[4]. Current guidelines for PA recommend that children and young people aged 5 to 18 years should engage in an average of at least 60 minutes of MVPA per day to elicit positive health outcomes[6].

Globally, physical inactivity is a major public health concern and efforts to increase overall PA and decrease sedentary time across the population are encouraged[7]. Recent European objectively-measured PA data suggests that the proportion of children meeting the PA guidelines ranges from just 2.0% to 14.7% in girls and 9.5% to 34.1% in boys[8]. Furthermore, survey level data from the latest Active Healthy Kids Wales Report Card within Wales, United Kingdom suggests that just 34% of children aged 3–17 years are meeting these guidelines[9]. In response to this data, the expert research group concluded the need to strengthen efforts in creating opportunities that increase children's PA. This group also highlight the gap in nationally representative data[9].

However, accurately measuring children's PA levels presents a number of methodological limitations[10]. Self-report methods including questionnaires are associated with subjectivity issues such as recall bias and are not advised in children younger than 10 due to their limited ability to accurately report PA[10]. On the other hand, whilst objective measurements such as accelerometry can measure PA across the domains of frequency, intensity and duration, they require participant adherence and are high-cost for researchers. Thus, as increasing levels of PA in childhood improves CRF and higher levels of CRF are associated with positive health outcomes[11], measuring CRF in children through methods such as the 20m shuttle run test (20m SRT) provides a valid, low-cost and pragmatic approach to assessing health-related PA interventions[12].

Children spend a significant amount of time in school and schools provide access to large populations from a range of socioeconomic backgrounds. With evidence demonstrating the rising levels of childhood physical inactivity, schools are targeted as a key setting to improve children's overall PA levels and health outcomes through implementing school-based running programmes[13]. Universal interventions directed at all children are attractive to schools due to their perceived lack of stigma, their ability to reach whole-classes and their potential in reducing health inequalities in later life[14]. Furthermore, teacher-led programmes that are low cost and require limited resources are favoured by schools in a time of education budget cuts and academic pressures. Comparable to other health behaviours, physical inactivity levels are higher amongst children from lower socio-economic groups[15]. Research has highlighted a scarcity of evidence examining child PA interventions across socio-economic groups[16].

However, the concern that intervention effects are stronger amongst children with better health outcomes as opposed to higher-risk children has been highlighted[17]. Thus, to avoid exacerbating the inequality paradox[18], it is important to examine the effects of universal school-based programmes across socio-economic groups.

With this said, an increasing number of running programmes are now available to schools [19], and in some cases are widely adopted despite limited evidence existing of their efficacy or effectiveness[20]. The Daily Mile (TDM) was established in Scotland in 2012 by a primary school headteacher to address concerns over pupils' perceived lack of CRF. This daily, teacher-led activity involves primary school children walking, jogging or running for 15 minutes during class-time within the school grounds[21]. The intervention's simple design and replicability has resulted in rapid uptake and is now being delivered in over 480 schools in Wales, and over 10000 schools worldwide[22]. This expansion was partly encouraged by rapid media and government attention, despite at the time no published evidence existing regarding its anecdotal benefits such as improved CRF, behaviour and concentration. Authors of a recent pilot study suggest that TDM is effective in increasing MVPA and CRF, decreasing sedentary time and improving body composition[23]. However, this study has been widely critiqued due to methodological weaknesses such as a small sample size. In a response, Daly-Smith *et al*[24] suggest a more cautious interpretation of these conclusions is required and call for further evidence of TDM in establishing an understanding of its impact, both positive and negative. Furthermore, a 'how to guide' has been published by the University of Stirling as an outline for schools regarding implementation and research findings[25].

The school environment is a complex system constructed of varying contextual factors[26]. This dynamic environment generates challenges for school-based interventions such as TDM to follow a uniform implementation method resulting in sustainability issues and a lack of evaluation[27]. Conflict also exists between the need for schools to strictly adhere to intervention design, recognised as intervention fidelity[28], and the variety of barriers and facilitators that influence the delivery and success of implementation such as adaptability and flexibility [29]. Previous research into school-based running programmes has demonstrated the variability in implementation across schools[30]. Interventions often lack foundation research assessing the acceptability and feasibility[31] which provide insights that inform future intervention implementation. In the case of TDM, the rapid adoption encouraged by media support and celebrity endorsement has resulted in wide global uptake at the detriment of feasibility studies assessing implementation factors. With evidence demonstrating that better quality implementation results in improved outcomes, this research is invaluable. Research has advocated that in order to interpret the evaluation of intervention outcomes, it is necessary to also examine the intervention components and implementation factors[28].

Two recent qualitative studies exploring the implementation processes and participants' experiences of TDM identified a number of factors associated with intervention success [32,33]. These included a need for simple core intervention components, flexible delivery encouraging teacher autonomy and intervention adaptability. Benefits cited by teachers included improved teacher-pupil relationships and the positive impact on pupils' health, well-being and fitness[33]. In contrast, a number of barriers were identified such as weather, resources and the perceived impact on learning time. Furthermore, the delivery style varied widely between schools, warranting further investigation into how delivery affects participants' experiences. These studies provide an important contribution to the understanding of implementation and experiences of TDM. However, both studies focussed solely on teachers involved in delivering TDM and the authors called for further research to incorporate children's views. To date, no research exists examining implementation factors of TDM from a whole-school perspective i.e. from pupils, teachers and senior management. In order to

develop and deliver effective interventions, it is vital to gain the viewpoint of the recipients of interventions; the pupils. In addition, it is important to understand the processes, barriers and facilitators of universal interventions from a whole-school perspective, incorporating objective measures of outcomes with qualitative research from a whole-school perspective of headteachers, teachers and pupils to improve understanding. This research is essential in informing the future delivery and sustainability of widely adopted interventions such as TDM.

The primary aim of this mixed-methods study was to explore the pupils', teachers' and headteachers' experiences of The Daily Mile and understand whether experience was related to implementation. The secondary aim of this study was to examine the association between The Daily Mile and children's cardiorespiratory fitness and compare this association between children in high and low socio-economic groups.

## Materials and methods

This mixed-methods study adopted a natural experiment approach with six primary schools interested in implementing TDM in south Wales, United Kingdom. Qualitative (headteacher and teacher 1:1 interviews, pupil focus groups) and quantitative measures (20m shuttle run test) were employed at two points (baseline and follow-up). Thematic analysis of qualitative measures was used to generate themes regarding the implementation of TDM and the associated experience of participants from a whole-school perspective. Multiple linear regression model analysis of quantitative measures was used to examine the effect of TDM on the CRF of children in high and low socio-economic groups.

### Ethics

Ethical approval was granted by the College of Medicine Research Ethics Committee (approval number 2017-0009A). Headteachers, teachers and parents provided informed written consent and children written and verbal assent prior to participating in the research study. All participants were reminded that their participation was voluntary and they had the right to withdraw from the research at any stage. All personal data such as school names and pupil names were anonymised. Paper based data (consent) was stored securely in a locked cupboard and electronic data (interview and focus group transcripts, quantitative data) was stored in password protected documents on a secure University server.

### Study design

This research study is a natural experiment with six schools who expressed an interest in implementing TDM. A natural experimental approach is considered the most suitable methodology when intervention implementation cannot be controlled by the researcher[34]. In the case of this research study, this was due to the rapid adoption of TDM encouraged by media and political attention[35]. In this research study, schools began delivering TDM at three time-points aligned with academic terms (School A–January 2017, start of spring term, School B–May 2017, start of summer term, School C-F–September/October 2017, start of autumn term). Data collection was completed in two phases to reflect the two academic years (Phase one 2016-17- School A and B, Phase two 2017–18 –School C-F). Data collection was conducted at two time-points; baseline (before implementation) and follow-up (3–6 months post implementation). A schematic diagram representing data collection periods across schools A-F is provided for clarity in the supporting information (S1 Appendix). A mixed-methods approach utilising both qualitative exploration and quantitative analysis was adopted to examine the research aims. Implementation level of TDM was not directly measured in this study but rather, emerged anecdotally through qualitative analysis. The purpose of this study was to

understand the experiences of participants in relation to implementation of TDM to inform future practice and sustainability rather than to develop new theory.

## Participants and setting

A convenience sample of six primary schools (School A-F) from south Wales, United Kingdom, who were about to implement TDM within their school agreed to take part in the research study. This sampling method was chosen with the aim to gather information-rich cases from schools committed to implementing TDM[36]. At the time of the study, there was political and public health support for primary schools within Wales to deliver TDM[35]. The schools participating in this research study were members of the HAPPEN Network (Health & Attainment of Pupils in a Primary Education Network), which aims to evaluate and share the evidence base for interventions currently delivered in primary schools in order to improve children's health, wellbeing and education outcomes[37].

The initial school recruitment process was facilitated through an Active Young People (AYP) Officer from the Local Authority's Sports Development team through an existing partnership with HAPPEN. The AYP officer had established links with all primary schools in their cluster area within the Local Authority and emailed these schools with an expression of interest in implementing TDM. Six primary schools (Schools A-F) responded and were subsequently contacted through HAPPEN via email regarding their intention to implement TDM. Recruited schools were then contacted via a telephone conversation with the headteacher. The percentage of pupils eligible for free school meals ranged from 4–54% for the six schools (national average 19%)[38]. The school size ranged from 175 to 275 pupils. Schools had minimal experience of implementing previous whole-school running programmes.

Following headteacher consent, the lead researcher (EM) delivered an information session about the study and distributed information sheets and consent forms to pupils aged 9 to 11 years (years 5–6) and their teachers at a school assembly. Each assembly provided pupils and teachers with the opportunity to ask questions about the research study. All pupils from years 5&6 from schools A-F were invited to participate in both the qualitative and quantitative measures. Pupils had the option to consent to participate in one or both measures in consent forms. Headteachers and all teachers from years 5&6 from the six schools were invited to participate in the qualitative measure.

## Instruments and procedures

Data collection was completed in two phases through the existing HAPPEN project, presented in the supporting information (S1 Appendix). Phase one (Schools A and B) baseline data collection was conducted in January (School A) and May 2017 (School B) and follow-up data collection was completed in July 2017. Phase two (Schools C, D, E, F) baseline and follow-up data collection was completed in September/October 2017 and March 2018. Both phases and time points followed identical protocols. Qualitative and quantitative assessments were carried out by trained researchers.

**Qualitative measures.** A qualitative approach is regarded the most suitable methodology in exploring barriers and facilitators of programme implementation[39]. In order to explore the primary aim of this research study, semi-structured 1:1 interviews with headteachers and teachers and focus groups with pupils were employed to gain an insight into implementation and experience of TDM in the primary school setting. This consisted of focus groups with pupils at baseline (n = 2) and follow-up (n = 4), 1:1 interviews with teachers at baseline (n = 3) and follow-up (n = 6) and 1:1 interviews with headteachers at follow-up (n = 2). A further breakdown of interviews and focus group participation by school can be found in the

supporting information (S2 Appendix). Interviews with headteachers and teachers were conducted by one researcher during the school day either by telephone or face to face on the school premises according to individual preference. Pupil focus groups were completed during the school day within a private room at the school setting, with two researchers present. The lead researcher (EM) was female and had previous experience in conducting interviews and focus groups with both adults and children in the field of school-based research. The researchers ensured that interviews and focus groups were conducted with minimal disruption to the school day and at a time that was convenient for teachers and pupils.

Each focus group was conducted by year group and consisted of between six and eight pupils[40] aged 9–11 years of mixed physical activity ability and gender. Class teachers were provided with a list of consented pupils and selected pupils fulfilling this criteria. Teachers were reminded of the need to include pupils of a range of physical activity abilities. This list was discarded following selection of pupils and a final list of pupils participating in focus groups was not recorded to ensure anonymity. All interviews and focus groups followed a semi-structured topic guide, initially developed by EM and CT and reviewed by SB to address the qualitative research aims. In order to explore participants' experiences of TDM, it is important to consider the barriers, facilitators and factors affecting sustainability. These factors are consistently included in other research evaluating school-based interventions, and therefore framed the topics guides for this study[33]. The use of semi-structured topic guides facilitated a deeper exploration of subjects and allowed topics to form naturally during the interview process[41]. These topic guides were not piloted prior to data collection but were based on previous school-based programme research through HAPPEN[42]. Example questions included "How do you feel about implementing the Daily Mile?" (teacher) and "Would you like to carry on with the Daily Mile and why?" (pupil). Full topic guides for interviews and focus groups can be found in the supporting information (S3 Appendix). The duration of interviews ranged between 5 and 21 minutes and focus groups between 23 and 48 minutes. The lead researcher (EM) facilitated the interview process, whilst the other researcher provided technical support (digitally recording) and made field notes on key responses. At the start of each interview and focus group, researchers reminded the participants of the study aims, guidelines on anonymity and confidentiality and encouraged participants' personal viewpoints. In order to achieve neutrality, researchers emphasised that they remained impartial and there were no right or wrong answers. In order to gain respondent validation, these notes were verbally summarised through member checking with interviewees at the end of each interview. To ensure trustworthiness, the researcher's interpretation of responses were summarised for corrections, clarification or confirmation by participants[43,44].

**Quantitative measure.** In order to examine the secondary aim of this research study, children's CRF was assessed using the 20m shuttle run test (20m SRT). The 20m SRT was conducted at the University's indoor athletics facilities and followed procedures outlined in the Eurofit Battery[45]. During this continuous running test, participants run between cones placed 20m apart in time with bleeps recorded on an audiotape. The initial running velocity of 8.5 km/h increases by 0.5 km/h each minute[46]. The time between consecutive bleeps decreases as the test progresses and the last shuttle a child is able to run is recorded. Cut points classifying children as fit and unfit were assigned according to total number of shuttles (fit: boys $> = 33$ shuttles, girls $> = 25$ shuttles) as these thresholds reflect cardiometabolic risk scores in children of this age group[47]. Prior to completing the 20m SRT, researchers provided verbal instruction about the test and a demonstration. Children were reminded of the study aims, their right to withdraw and provided additional verbal consent prior to participating.

## Statistical analysis

**Qualitative analysis.** The qualitative component of this research study adopted an interpretive approach through thematic analysis in order to gain an understanding of participants' experiences of implementing TDM. All interviews and focus groups were digitally recorded and transcribed verbatim in Microsoft Word. The process of analysing the interview and focus group data followed the steps outlined by Burnard (1991)[48]. To begin, each transcript was independently read several times by two researchers (EM and CT) to facilitate immersion in the data. The researchers (EM and CT) then followed an independent open coding process to allow participants' views to be summarized by assigning words or phrases to quotes or paragraphs. This initial list of freely generated categories following review of the transcripts aimed to encapsulate interviewees' responses and were subsequently grouped according to the overarching theme. Through this process, broader categories were combined to produce one higher-order heading that captured the overall meaning of responses. This process was repeated whereby similar categories were synthesised to produce a final list of themes and subthemes. Both researchers (EM and CT) compared their lists of themes and sub-themes to ensure accuracy and consistency. If there was a discrepancy or disagreement in coding, a third researcher (SB) adjudicated. This method enhances the validity of categories assigned and attempts to reduce researcher bias[48]. The written notes taken on the day of the interview or focus group were compared with these topics to ensure an accurate account of participants' responses. Following this, the two researchers worked together through an extensive process to discuss codes and categorise them under final themes and sub-themes (S4 Appendix). The lead researcher (EM) then manually worked through each transcript and coded the responses according to the final list of themes and sub-themes. All responses grouped by themes and sub-themes were compiled to a master copy document that was used for reference to write up the findings.

**Quantitative analysis.** Analyses were performed in STATA (version 15). Multiple linear model regression analyses was used to examine the association of TDM on children's CRF. Schools provided date of birth (to calculate age) and postcodes (to calculate individual level deprivation) for consented pupils. Discrepancies in numbers within results tables are due to missing age and postcode data. The explanatory variable (individual pupil deprivation) was adjusted for confounders (baseline age, gender). Analyses were also clustered by school to account for school-level differences. Deprivation was assigned as an area-based socio-economic measure using the Welsh Index of Multiple Deprivation (WIMD)[49]. Weighted scores for eight domains of deprivation are calculated as a WIMD score for each LSOA. WIMD scores are ranked from most to least deprived and grouped into quintiles (1 = most deprived, 5 = least deprived). For the purpose of this study, a binary *deprived* (WIMD quintiles 1, 2) and *non-deprived* (WIMD quintiles 3, 4, 5) variable was assigned representing low and high socio-economic groups.

A constraint of school-based research is the potential for missing data due to pupil absentee at random, through illness or other school commitments that prevent them from participating in data collection, contributing to bias in results [50]. To overcome this, missing data in this sample were imputed. The Multivariate Imputation by Chained Equation (MICE) method in STATA using baseline and follow-up data (shuttles, age, deprivation) was used to impute missing data for those missing at either time-point. Data were assumed to be missing at random (e.g. probability of being missing does not depend on the missing value) on the basis that there was no significant difference of baseline shuttles between groups (missing at follow up, present at follow up).

## Results

### Qualitative results

The primary aim of this research study was to explore the pupils', teachers' and headteachers' experiences of TDM and understand whether experience was related to implementation. The overall implementation of TDM varied widely amongst schools. Although this was not measured directly, this variation in delivery styles emerged from the transcripts and is reflected in the overall experiences of participants. Two over-arching themes arose from the data; 1) The Daily Mile implementation and 2) impact on learning, health and wellbeing. Theme one, The Daily Mile implementation will be discussed in relation to the conflicting sub-themes that reflect the varying implementation and experience of participants; flexible vs rigid principles, curriculum time vs playtime, competitive vs non-competitive, active teachers vs passive teachers, supported vs unsupported, and summer vs winter. Theme two, impact on learning, health and wellbeing will be discussed through the following sub-themes; behaviour and concentration, physical activity and sport, psychological benefits, social benefits.

**The Daily Mile implementation.**

<u>Flexible vs rigid principles</u>. This theme relates to the varying implementation style adopted by schools; either demonstrating flexibility and adaptability or following the original principles set out by TDM. School A suggested that it required flexibility from individual classes within the school and implementation reflected this;

> *Yes, different classes do different things, so what works for one class doesn't necessarily work for another class. In Year 2, we tend to run it about two o'clock in the afternoon because we don't get an afternoon play, so we do it then and it breaks the afternoon up, which is quite nice. Other classes have been doing it first thing in the morning, other classes have done it last thing, quarter past three, different things work for different classes.* (Teacher, School A, Follow-up, T12)

This view was mirrored by school D who supported an 'anything is better than nothing' perspective;

> *I think it is do what suits you. Don't worry about what anyone else is doing. Even if it is 5 minutes it's 5 minutes it is better than nothing. I just think just to think carefully about it is worth the infants doing it and then just finding that time slot really, try looking a bit differently at your school time table to free up a bit of time that is maybe non-productive time at the moment.* (Headteacher, School D, Follow-up, T3)

In addition, implementation that encouraged any form of movement as opposed to enforcing running was discussed positively by both teachers and pupils in relation to the importance of participation;

> *One thing about it is that you're meant to try and do your best in it, do what you can, like run it, walk it, or jog it, doesn't really matter, as long as you actually do it, as long as you do it.* (Pupil, School E, Follow-up, T14)

> *There's a couple of, a couple of girls I'm thinking of who I think they've just skipped [laughs] about, I don't think they've done any running, they've skipped the mile every day, but at least they were doing it.* (Teacher, School A, Follow-up, T12)

In comparison, school B believed maintaining a consistent implementation method based on the original Daily Mile principles was important;

*What happens with these initiatives is they get put into school and then the school change them, so then it doesn't stick really to the principles of TDM which were there in the first place it doesn't really fit with the core principles that TDM set out.* (Teacher, School B, Follow-up, T10)

In order to ensure that schools do not deviate from these principles, this teacher suggested an external Daily Mile advisor for schools;

*So I think that there needs to be somebody, an advisor that schools can go to to make sure that they are sticking to their original Daily Mile plan and not turning it into something else. . .it just takes away from the whole point of it . . . there are 10 core principles set out in TDM documentation, I just think it needs to be you know, stuck to that original plan.* (Teacher, School B, Follow-up, T10)

However, a consistent implementation method that maintained the original principles was associated with a lack of enthusiasm and engagement by many participants;

*If the children are doing something like that every day and it's the same thing day-in-day-out I feel they. . .the novelty wears off . . . I think it's like a lot of initiatives, you know, when we first have these initiatives the children are all up and ready and keen and love doing these things, but unless it changes or unless something is added to it, or unless they get something out of it, it's just a day-in-day-out thing.* (Teacher, School A, Follow-up, T6)

A rigid implementation of TDM and lacking variety disengaged pupils. In order to maintain enthusiasm, pupils incorporated elements of play;

*I don't mind it now because I try and mix play and TDM together somehow. We do like tapping on the head while run around the yard.* (Pupil, School F, Follow-up, T16)

In addition, pupils suggested a number of ways in sustaining their engagement such as utilising sports equipment and varying the location;

*I also think that they could put sports equipment in the middle because some people might want to do a different sport and not running.* (Pupil, School F, Follow-up, T16)

*We could change it by going like a different place, not just the same place because it's quite boring if you just go round the same place* (Pupil, School E, Follow-up, T13)

Therefore, incorporating flexibility lead to a more positive experience and increased engagement from most pupils.

<u>Curriculum time vs playtime.</u> There were conflicting approaches adopted by schools with regards to the time allocated for TDM. This comprised of either implementation during the taught curriculum or an additional/replacement of pupils' playtime. One of the main concerns for all headteachers and teachers was the curriculum pressure by educational inspectorates. Although schools wanted to implement TDM, finding 15 minutes within the narrow curriculum was a challenge;

*We were all sort of a bit sceptical when it first come out, sort of just timings it is in school, it's not that we didn't feel it would be a good thing to do, it's just we've got so much to do in school*

*it was sort of timing that was our issue as a class teacher, to fit everything in and to do it not to take up their playtime.* (Teacher, School B, Baseline, T9)

For some schools, this curriculum pressure forced them to find alternative opportunities in the day to deliver TDM. The headteacher from school F explained that curriculum and parental pressure influenced their decision in replacing pupils' playtime;

*There are so many pressures on schools these days with their drilling with welsh and this that and the other it is so difficult. We have tried to get it that it doesn't eat into lesson time. I think there were some concerns with parents with regard to would this 15 minutes eat into lesson time which is why we tried to put it into playtime.* (Headteacher, School F, Follow-up, T15)

For many pupils, the replacement of their playtime was a negative factor associated with their experience;

*If it wasn't taking up our play time which is one of the fun moments of the day, then I would do it, because it is during play I don't really want to do it.* (Pupil, School F, Follow-up, T16)

However, this headteacher recognised pupils' frustration and utilized their pupil voice group to incorporate pupils' suggestions and maintain engagement;

*One thing that has gone well, we have got a portable speaker so we put music on for the children to run around to music, they quite like that. One suggestion. . .we have a 'Healthy Pupil Voice' group, is that they are going to create different playlists to try and put a bit of variety in it.* (Headteacher, School F, Follow-up, T15)

In contrast, schools without an afternoon playtime coordinated TDM to be delivered as an additional playtime through a restructure of the school day;

*We thought there is no point in doing it in the morning because it is usually afternoon when they dip . . . So we actually changed the lunch hour, we shaved 10 minutes off the junior lunch hour, so they go back into class 10 minutes earlier, so that means they were not missing any of their lesson time. So we have actually put the 10 minutes into the afternoon break and they seem to find that this doesn't bother them at all . . .I think everyone enjoys the 10 minutes of fresh air and the break. They all go back a bit more replenished and bit ready for the next hour challenge.* (Headteacher, School D, Follow-up, T3)

The addition of an afternoon playtime to participate in TDM was supported by all pupils;

*All the infants [younger Key Stage] get their play, they put TDM in as basically our third play, which is good.* (Pupil, School D, Baseline, T1)

With curriculum pressure being highlighted by schools, some overcame this by integrating TDM within Physical Education (PE) lessons;

*We looked at the time tables for everybody and realised that was the only spot that we had. But what we tend to do is if the key stage 2 staff have PE on a particular day they won't do*

*their Daily Mile at 12.50 they will do it during their PE session. So they tag it on twice at the beginning of the PE session or the end.* (Teacher, School C, Follow-up, T11)

In addition, school B used TDM in achieving the weekly recommended guidelines for PE provision;

*We work quite smartly here so we link, we try to link everything in as best as we can, as I said fits in with my topic. . .That's not the same with every class but, you know, there are opportunities to link it with curriculum, yeah, with curriculum target skills, so that's good. And it will help to count towards our overall PE time for the week as well.* (Teacher, School B, Baseline, T7)

Overall, the addition or replacement of pupils' playtime for TDM was a significant contributing factor to pupils' experience.

Delivering TDM as an extra afternoon playtime and an additional break from lessons was a positive factor influencing pupils' experience. However, replacing scheduled play caused a significant problem for pupils who enjoyed the autonomy and freedom of playtime. Although TDM was not intended to act as a replacement to PE, for some schools this was the only opportunity in the school day that did not take away from curriculum time.

<u>Competitive vs non-competitive</u>. Conflicting messages regarding competition within TDM were conveyed by participants. Teachers discussed the challenge that existed in balancing competition. For some pupils, the competitive element fostered participation and enthusiasm whereas for others, competition completely disengaged them. This teacher felt that children thrived in competitive environments, but this was at odds with the original Daily Mile principles;

*I think they like the competitive element, which is not what it's meant to be. Then some of them, they're just not enthusiastic for sport and that's the negative isn't it? We've got the ones who are very enthusiastic and then the ones who really can't be bothered. . .And I know it's not meant to be competitive but that's the sort of, that's what children like. They like to do their best. They like to win. So it's difficult.* (Teacher, School B, Baseline, T5)

However, finishing last was a cause of concern for some pupils who associated this with ability, suggesting that a continuous bout of 15 minutes of activity was favoured than the completion of an actual mile;

*Well some of them don't like just running in general, but some of them, and some of them are desperate not to be last, like no-one wants to be the worst runner in the class.* (Pupil, School E, Follow-up, T14)

Many teachers also recognised the need for rewarding pupils to encourage their participation, stating that the wider benefits of participation were not valued by pupils;

*I know it's not meant to be a competitive thing but there needs to be some sort of reward. They need to see some sort of purpose in doing it. Yeah okay, I'm doing it to see if it affects my performance in school, but that doesn't mean anything to our children.* (Teacher, School B, Baseline, T5)

To overcome this challenge, another teacher from this school suggested incorporating goal setting and enabling competition through personal targets;

*On Healthy Schools week I did for Year 6 because it's Healthy Schools week so we did record times for one week as the children were not against each other, but they were recording their own personal best and that was the week that really worked well with my class particularly, just because they were, not being competitive with anybody else, but they were setting a target for themselves, do you know what I mean*? (Teacher, School B, Follow-up, T6)

Overall, implementation that incorporated goal setting as a means of highlighting progress and personal achievement fostered engagement and motivation to participate in TDM. Findings regarding competition were mixed and depended on the individual pupil and their perception of competition, either thriving or disengaging in this environment.

Active teachers vs passive teachers. The involvement and role of teachers during TDM was discussed by participants. Two clear themes emerged; active teacher involvement and participation or passive and disengaged teachers. Participants discussed the positive effects of teacher involvement in the implementation of TDM. For pupils, teacher encouragement and participation influenced their engagement;

*They [teachers] actually try and like do it, they would tell us and in encouraging and inspiring way. They're like, come on, come on, go on, you can do it, come on!* (Pupil, School D, Baseline, T2)

Teachers were also aware of the enabling role they played in pupils' participation and supporting children that found TDM challenging. In addition, this teacher also acknowledged the benefit of participation on teachers' fitness;

*Staff are good, I am trying to encourage them to run it at the same time as the children. Some members of staff will run it as well; I like to join in as well. Even if it's a case of just walking around with them for those children who are struggling. I am keen that the staff don't just stand around watching them, that they try and get involved as much as possible for our own fitness levels as well . . .I think I would encourage other schools to take it on and try to keep it up. To try to get the staff more involved. You get the staff to encourage the children to run and obviously set a good example by them doing as well.* (Teacher, School C, Follow-up, T11)

In contrast, some pupils from other schools discussed the passive involvement of teachers and the negative effect this had on participation. For pupils, disengaged teachers resulted in disengaged pupils and rule breaking;

*I think that the teachers should start running it, because they're just like standing there while we're doing all the running and I feel like they should be doing it. . .If they joined in I would run more.* (Pupil, School E, Follow-up, T14)

*I think that the teacher should actually watch because everyone usually cheats, [teacher] is usually just on her phone [all laugh].* (Pupil, School E, Follow-up, T14)

The importance of role modelling was also recognised by teachers in reference to the correlation between the lack of teacher enthusiasm and pupil engagement;

*I just get the feeling that they're [teachers] really not that into it so they haven't then passed on their enthusiasm to the children and I think that staff have been supervising the children but not joining in, and then it becomes something that children are being told to do instead of something children and staff are doing together.* (Teacher, School B, Follow-up, T10)

Teachers seeing value and benefit to pupils was crucial to gaining their support and enthusing pupils. However, some schools discussed the conflict that existed between the engagement of teachers from different year groups;

*I think some can definitely see the benefit of it, others I think feel that it is something else, another initiative, and that's probably been the difference between Foundation phase [ages 4–7] and Key Stage 2 [ages 7–11] as well is that the Foundation phase staff have been quite enthusiastic about it, Key stage 2 have been a little bit apathetic if you towards it and if it doesn't come with enthusiasm from the staff the children will pick up on that won't they*? (Headteacher, School F, Follow-up, T15)

This view was reflected by school B who felt that TDM favoured the lower key stages in which the curriculum is delivered through play;

*The foundation stage staff are more engaged I think because it's easier to fit in their daily routine, because the children there are learning through play anyway. It's not a big chunk out of the curriculum when that is something that they [foundation phase] do anyway, and I definitely have better engagement from foundation stage staff, even at the idea stage than I had from Key Stage 2 staff.* (Teacher, School B, Follow-up, T10)

Teacher buy-in and active participation in TDM was important in modelling behaviour and motivating pupils. However, concerns were raised by some participants regarding the engagement of teachers from the higher key stage of primary school.

<u>Supported vs unsupported.</u> The varying level of support from staff, parents, stakeholders and the wider community were discussed by many participants. The importance of headteacher support was identified as a critical factor in gaining the support of parents. Communication through social media facilitated this;

*Initially the first week we had a few grumbles on Facebook [by parents] on the social side, basically complaining that their children were feeling sick after doing the run. . .But our Head, was quick to reply to that message and basically said that we were out and none of them complained about feeling unwell so, we think it is the parents job to try and support the school in new initiatives and that we are doing it for their children's health and wellbeing, so stop moaning about it basically and we haven't have any problems since.* (Teacher, School C, Follow-up, T11)

In addition to parental support, engaging with sporting role models within the community acted as a way of inspiring pupils about physical activity and fitness;

*We've had [professional footballer] and another [football team] player came in to talk about, to run with the children, and talk about the importance of running and fitness. And we also had the, the physiotherapist, the physio from [football team], he came in as well. So they've had a lot of encouragement and from parents and from the community really.* (Teacher, School A, Follow-up, T12)

However, a lack of wider support and the difficulty in maintaining children's engagement with TDM was discussed as a barrier to sustainability by other schools;

*No, we've not seen anybody, I'm just chatting to my colleague, nobody's come to join in, I know it's difficult but I think we probably could have done with a bit of support, and not just a one-off, somebody turning up for a day and saying, "Come on children," because I'm just speaking, I'm the deputy head here, I'm speaking on behalf of my junior staff here, we have found it quite a challenge ourselves then really day-in-day-out to get children doing something that they don't all want to be doing.* (Teacher, School B, Follow-up, T6)

Linking to discussions of the varying engagement by teachers of different years, this school believed that external support was necessary for the older key stages, highlighting a decline in local authority support;

*No, and we were supposed to [have support], initially there was a lady from [local authority sport team] the, is it the [local authority initiative]? Yeah they were supposed to be involved but she dropped off after the first lot of data collection and never came back so I think that might have made a difference. . . I think in the upper end of the school definitely, you know in the Key Stage 2 [year 3–6] I just think that they'll need that extra support I think.* (Teacher, School B, Follow-up, T10)

Therefore, supporting schools with the implementation of TDM would encourage sustainability. This includes support from parents and the wider community and backing from local authority sport and health teams through the provision of staff members dedicated to Daily Mile implementation.

Summer vs winter. Some teachers questioned the seasonal effects of TDM for pupils, speculating that pupils' enthusiasm was dependent on the weather;

*To be honest, with the weather I'm not sure whether it would have the same effect in the spring and autumn term, like it was the summer term which was fantastic term to do it. We could certainly give it a go, but I'm not sure it would have the same effect.* (Teacher, School A, Follow-up, T12)

However, despite summer being seen as an ideal term to implement, hot weather also created additional concerns around health and safety for a few teachers;

*The only problem I can really think of is when the temperature was very hot, you know, just making sure that they were hydrated and that we were, you know, they weren't too tired running in the heat and they weren't exhausted.* (Teacher, School A, Follow-up, T12)

The pressure of parental concerns regarding weather and safety were also highlighted, with parents expressing apprehension about their child engaging in physical activity in the heat;

*I don't know how it would, what would happen in the winter, like you know, and we had parents complain actually, "Don't let them run in the heat," so I can't imagine what would happen in the winter.* (Teacher, School B, Follow-up, T6)

Wet weather posed a problem in relation to clothing, finding alternative opportunities for physical activity (teachers) and safety concerns (pupils);

*When it is like drizzling, raining, a little bit, we still have to go out and sometimes the yard is really slippery and I have seen a lot of people falling over doing TDM when it has been raining because it is winter now and the terrible weather came and we are still doing it and lots of people are falling over and hurting themselves.* (Pupil, School F, Follow-up, T16)

This weather also created a practical barrier regarding clothing. Although TDM does not require specific clothing for implementation, reliance on school uniform posed a problem to schools. This included inappropriate footwear and issues related to hygiene, requiring schools to consider ways of overcoming this barrier;

*We have been out there on days that are drizzly but the children slip and slide on the grass and then if you do it on the yard, they, our yard tends to puddle so they're coming in with wet shoes. We have to look into really bringing in separate shoes for running* (Teacher, School A, Follow-up, T12)

**Impact on learning, health and wellbeing.** The majority of participants discussed the impact of TDM on pupils' learning, health and wellbeing. Conflict existed between participants' perceptions of the impact on behaviour and concentration. However, discussions on health and wellbeing were generally positive, covering the physical, psychological and social domains.

Behaviour and concentration. Participants' views of the effect of TDM on the subsequent behaviour and concentration of pupils were mixed. Some teachers observed improvements in pupils' behaviour and concentration following participation in TDM.

*Just generally they have been much better, calmer coming into class in the afternoon because of it. They are coming in and ready to start working and that has been great, we have noticed a big difference there. In terms of concentration levels as well, they seem a bit more perkier. . .I do think it has a positive effect and I think it has had a positive effect on energy levels and that knocks on into class time then. We've seen an improvement in behaviour in class definitely.* (Teacher, School C, Follow-up, T11)

This finding was also reflected by some pupils who felt that participating in TDM resulted in more efficient class work;

*Well, I wouldn't say it affects me but like, normally, I would just be really, like just doing my work, but now that I've done TDM it gives me a bit of a boost and now I'm starting to do my work a bit quicker.* (Pupil, School D, Baseline, T1)

This was supported by a teacher from school D who discussed this in relation to the theme of curriculum time vs playtime. The introduction of an additional afternoon break to participate in TDM was a conscious decision by this school who observed the positive impact on pupils' concentration;

*We have noticed an improvement in fitness and concentration. Our lunch time goes into 12.50pm which starts the first lesson, so by 2.00pm the children are flagging. So to have the Daily Mile is a god send at the moment. When they come back in after their 10–15 minutes because the time is going less and less the more they are doing it, there are ready for their last lesson then. Whereas before it would have been so difficult to teach that last lesson.* (Teacher, School D, Follow up, T4)

However, some pupils suggested that the perceived improvements in behaviour resulted from a reduction in energy levels following participation;

*I think it's improving the behaviour a bit because people don't have as much energy to mess* around. (Pupil, School D, Baseline, T2)

In addition, pupils highlighted a negative association between energy levels and concentration;

*I think there might be one [problem] when it gets you, when it gets you so tired when you go back into class, that you lose all your concentration until you regain it* (Pupil, Baseline, School D, T2)

In contrast, other teachers raised the challenge of settling pupils back into lessons following TDM, who felt that pupils returned to class over-excited;

*But we have found to do the run and come back into class has had the opposite to the desired effect, that it didn't settle them, it just made them more hyper.* (Teacher, School B, Follow-up, T6)

Overall, the effect of TDM on behaviour and concentration is largely dependent on the individual pupil. For those reporting positive improvements, longer-term sustainability is likely to be encouraged in order to foster school-wide benefits to learning. However, the conflicting statements of over-excitement and tiredness that result in a negative impact on learning are likely to discourage schools from sustaining TDM.

Physical activity and sport. Many pupils believed participating in TDM improved their attitude towards physical activity;

*I actually think it's had an effect on me because it actually gets me going when . . . Well if I'm not doing very well in school it's usually PE day so then we just go out and do PE so it like gives me positive attitude to my learning and physical activity.* (Pupil, School E, Follow-up, T13)

Teachers also discussed pupils' positive attitudes towards physical activity and elements of behaviour change;

*Their attitude towards fitness has improved and what we have noticed also is that most of them after Christmas have come back to school with fitbits. They are tracking their steps now, so hopefully that will have a long term effect on them you know.* (Teacher, Follow-up, School D, T4)

Pupils were positive about the additional opportunities to be physically active and the contribution towards structured sports participation;

*I thought it was a good thing, I don't really get to do much running at home.* (Pupil, School F, Follow-up, T16)

*Very much. I really like TDM because it keeps me active and it like helps me like to train in the week for my football and all that.* (Pupil, School D, Baseline, T1)

Furthermore, some pupils and teachers attributed improvements in pupils' sporting achievements to participating in TDM;

*It has changed the way like girls only like to do gymnastics is has changed the way of my gymnastics skills are getting better and better. But then it is not just gymnastics there is also other sports like tennis, hockey and football sometimes. All types of sports are getting easier for lots of the girls and boys because they do TDM.* (Pupil, School F, Follow-up, T16)

*We've been in sort of local competitions and we've brought medals home from there and, you know, the children are saying 'oh it's because we've been practising every day' and they're putting it down to practice makes perfect which is quite nice as well.* (Teacher, School A, Follow-up, T12)

<u>Psychological benefits.</u> Many pupils acknowledged the associations between physical activity and wellbeing;

*So we get, even though we got active, our brains are getting healthier, so it's better for our minds as well, in the work after TDM.* (Pupil, School D, Baseline, T1)

Both pupils and teachers commented on feelings of happiness;

*Um, yeah, I think so because sometimes some of my friends are a little bit tired and angry in the morning and then when they do TDM they're kind of happy and stuff.* (Pupil, School E, Follow-up, T13)

*Well, happy children learn, so if they're happy they're going to learn, and they're certainly happy after running TDM!* (Teacher, School A, Follow-up, T12)

In addition, feelings of improved self-esteem and school competency were reported;

*Um, yeah, it makes me more confident, so I do better in school.* (Pupil, School E, Follow-up, T13)

Regular participation in TDM acted as a stress relief and in alleviating the pressures of exams at this Key Stage;

*Um, I think it just takes your mind off things and it really just helps.* (Pupil, School E, Follow-up, T13)

*I actually think it's better doing it as Key Stage 2 because you get more support and so it keeps you with a positive mindset.* (Pupil, School E, Follow-up, T13)

<u>Social benefits.</u> Many pupils reported a number of social benefits to participating in TDM. This included the opportunity to interact with peers and the positive subsequent effect during lessons;

*You also get to chill, at the same time talk to your friends which stops you from wanting to talk to your friends in lesson, since you've talked to them, you know what they want to say to you, they know what you want to say to them, so you don't really need to talk to them.* (Pupil, School D, Baseline, T2)

Teachers also reported the social improvements through group participation such as teamwork and cooperation;

*I'm thinking of a couple of girls in my class now because they're doing it together and I said, well go together, encourage each other, they are enjoying it a bit more now and they get out and they're all bringing their water bottles in and things like that now, so they are enjoying it more than the first sort of day when I said right, well you've got to go out, it's not just seven laps, you've got to keep moving for 15 minutes and you could see some grumbles, they're a lot more positive the more we're doing it and it's becoming routine.* (Teacher, School B, Baseline, T9)

*And not only has sort of their health and fitness improved, but their social skills have improved as well, because they were doing it together, yeah, they really loved it.* (Teacher, School A, Follow-up, T12)

Therefore, these positive discussions covering the physical, psychological and social domains are likely to encourage schools to continue delivering TDM in order to elicit the range of benefits observed on pupils' health and wellbeing.

## Quantitative results

The secondary aim of this research study was to examine the association between TDM and children's CRF and given the universal nature of TDM, compare this association between children in high and low socio-economic groups. Table 1 presents the descriptive characteristics of those that participated in CRF tests and the total sample (including imputed data). There was a total of 336 pupils in years 5 and 6 attending the six primary schools in this study. From this sample of eligible pupils, 229 pupils (68%) participated in the 20m SRT at baseline and 235 pupils (70%) at follow up. In total, 204 pupils (61%) completed the 20m SRT at both time points. The MICE imputation method utilising shuttles, age and deprivation accounted for an additional 34 pupils at baseline and 28 pupils at follow up. There was no significant difference ($p = 0.33$) between the mean number of baseline shuttles for those that participated in the 20m SRT and the total sample (including imputed). Results described below will be discussed in relation to imputed data.

The descriptive characteristics for shuttles and CRF at baseline and follow up (overall, deprived, non-deprived) are presented in Table 2. At baseline, 51% of participants were classified as fit. Overall, participants in the deprived group performed a lower number of shuttles in the 20m SRT compared to children in the non-deprived group at baseline (deprived:

**Table 1. Descriptive characteristics (participated in 20m SRT, total sample).**

| | Baseline | | Follow-up | |
|---|---|---|---|---|
| | **Participated in baseline 20m SRT** | **Total sample baseline 20m SRT (imputed)** | **Participated in follow-up 20m SRT** | **Total sample follow-up 20m SRT (imputed)** |
| **Age (years–at time point)** | 10.2 ± 1.0 (220) | 10.2 ± 0.9 (254) | 10.6 ± 0.6 (227) | 10.6 ± 0.6 (255) |
| **Boys** | 52% (117) | 54% (141) | 56% (130) | 54% (141) |
| **Deprived (WIMD quintiles 1, 2)** | 36% (79) | 36% (94) | 37% (86) | 36% (94) |
| **Shuttles (mean)** | 30.7 ± 19.3 (229) | 30.9 ± 18.5 (263) | 35.5 ± 20.5 (235) | 35.7 ± 19.8 (263) |
| **% fit** | 49% (110) | 49% (128) | 58% (135) | 60% (157) |
| **% fit (boy)** | 53% (62) | 54% (76) | 55% (72) | 58% (82) |
| **% fit (girl)** | 44% (48) | 44% (52) | 62% (63) | 63% (75) |

Mean ± SD (n); % (n)

**Table 2. Descriptive characteristics (overall, deprived, non-deprived).**

| | Overall | Deprived (WIMD quintiles 1 & 2) | Non-deprived (WIMD quintiles 3, 4, 5) |
|---|---|---|---|
| **Shuttles difference (baseline–follow-up)** | 5.4 ± 12.8 (204) (95% CI: 3.6 to 7.2) | 4.4 ± 12.2 (72) (95% CI: 1.5 to 7.3) | 5.7 ± 13.2 (127) (95% CI: 3.4 to 8.0) |
| **Shuttles difference imputed (baseline–follow** | 4.9 ± 15.0 (263) (95% CI: 3.1 to 6.7) | 4.7 ± 13.4 (94) (95% CI: 2.0 to 7.4) | 4.8 ± 16.0 (164) (95% CI: 2.3 to 7.3) |
| **Shuttles baseline** | 30.7 ± 19.3 (229) | 23.8 ± 17.1 (79) | 34.8 ± 19.6 (145) |
| **Shuttles follow up** | 35.5 ± 20.5 (235) | 28.4 ± 18.3 (86) | 39.8 ± 20.9 (144) |
| **Shuttles baseline imputed** | 30.9 ± 18.5 (263) | 23.7 ± 16.0 (94) | 35.2 ± 18.7 (164) |
| **Shuttles follow-up imputed** | 35.7 ± 19.8 (263) | 28.4 ± 17.9 (94) | 40.0 ± 19.8 (164) |
| **% fit (baseline)** | 49% (110) | 32% (25) | 59% (84) |
| **% fit (follow up)** | 58% (135) | 43% (37) | 67% (94) |
| **% fit imputed (baseline)** | 51% (132) | 30% (28) | 62% (99) |
| **% fit imputed (follow up)** | 60% (157) | 44% (41) | 70% (112) |

Mean ± SD (n); 95% CI = 95% confidence interval; % (n)

23.7 ± 16.0, non-deprived: 35.2 ± 18.7) and follow-up (deprived: 28.4 ± 17.9, non-deprived: 39.8 ± 20.9). A lower proportion of participants in the deprived group compared to the non-deprived group were classified as fit at baseline (deprived: 30%, non-deprived: 62%) and follow up (deprived: 44%, non-deprived: 70%). Both groups demonstrated equal increases in shuttles between baseline and follow-up (deprived: 4.7 ± 13.4, non-deprived: 4.8 ± 16.0). However, these results exhibit large standard deviation and wide 95% confidence intervals (deprived: 2.0 to 7.4, non-deprived: 2.3 to 7.3), demonstrating the variability that is present among this sample. A further breakdown of the descriptive characteristics for shuttles and CRF of the sample categorised by school (A-F) can be found in the supporting information (S5 Appendix). Using regression analysis to adjust for age and gender showed there was no significant difference in the increase in shuttles run for deprived compared to non-deprived children (Tables 3 and 4).

## Discussion

Schools are considered a key setting in combating the rising levels of childhood physical inactivity through implementing universal running programmes aimed at increasing children's PA levels and CRF. However, their simple design and widely scalable nature with limited resources and low-cost has resulted in widespread adoption lacking evaluation of both quantitative outcomes and qualitative implementation factors that ensure success and sustainability. To date, TDM has been implemented in thousands of schools globally with the aim of improving children's PA, CRF, health and wellbeing[51]. However, limited research exists examining the implementation and experience of TDM from a whole-school perspective[32,33]. Given its rapid expansion, this research is invaluable in providing schools with an evidence-based approach to successful implementation. Therefore, the primary aim of this study was to

**Table 3. Regression model 1—Difference in shuttles baseline to follow-up, baseline age, boy, deprived, clustered by school.**

| | Coef. | P>|t| | 95% Confidence Interval | |
|---|---|---|---|---|
| **Decimal age** | .56 | 0.55 | -1.71 | 2.84 |
| **Boy** | -2.62 | 0.16 | -6.67 | 1.43 |
| **Deprived** | -1.49 | 0.56 | -7.64 | 4.66 |
| **_cons** | 1.38 | 0.16 | -21.31 | 24.08 |

**Table 4. Regression model 2—Difference in shuttles baseline to follow-up imputed, baseline age, boy, deprived, clustered by school.**

|  | Coef. | P>|t| | 95% Confidence Interval | |
|---|---|---|---|---|
| **Decimal age** | .23 | 0.76 | -1.56 | 2.02 |
| **Boy** | -2.45 | 0.37 | -8.85 | 3.95 |
| **Deprived** | -0.05 | 0.99 | -4.89 | 4.78 |
| **_cons** | 3.60 | 0.62 | -13.95 | 21.15 |

explore pupils', teachers' and headteachers' experiences of TDM and understand whether experience was related to implementation. Findings from this study identified a variety of implementation factors that affected participants' experience which will be discussed and summarised to provide a set of recommendations to schools.

Headteachers, teachers and pupils discussed a range of factors associated with implementation in relation to the experience and engagement with TDM. These barriers and facilitators to effective implementation identified by participants are consistent with recent research into school-based running programmes[30,33]. Implementation in this study, as captured through interviews and focus groups varied widely amongst schools and is reflected in the contrasting themes that emerged from the data. However, implementation was not directly measured and future research into TDM would benefit from examining the strength of outcomes in relation to implementation level and style. Conflict existed between schools on how TDM should be delivered, raising the issue of fidelity to the intervention. Some teachers felt following the original principles was essential. However, others and in particular the pupils advocated for flexibility, cited as a facilitator to implementation in the literature[28]. The contextual differences that schools face contribute to the challenge in the implementation of interventions following a uniform method. Previous qualitative research exploring implementation of TDM highlighted the importance of flexible implementation in facilitating teacher autonomy and engagement [32,33]. Furthermore, variation in implementation and flexibility has been documented as a facilitator in other school-based running programmes[30]. With flexibility consistently identified as a key factor to the effective implementation of interventions, it is essential for future programmes to be designed with this at the core. These findings highlight the importance of designing adaptable school-based programmes to fit within the varying contexts of schools, rather than a 'one size fits all' model. However from a research perspective, variation in delivery and fidelity to the original intervention design poses a number of challenges for evaluating school-based interventions.

This is of particular importance as one of the most significant barriers raised by pupils was a lack of variety. After the initial excitement of a new school activity, pupils commented on feeling bored and lacking enjoyment, impacting on pupils' participation and the longer-term sustainability of TDM. Indeed, research has suggested that the novelty of unique PA methodologies may only elicit increases in activity in the short-term, with original behaviours returning as motivation for participation decreases[38]. As TDM has no defined length, adherence to longer-term implementation may require additional techniques to encourage behaviour change that maintains motivation[39]. One such suggestion is to target pupils' enjoyment to facilitate involvement with interventions[38]. One school addressed this challenge by utilising their 'pupil voice group' to suggest alternative methods for implementation. Pupils from this school proposed incorporating music and discussed the positive effect this had on engagement. This highlights the importance of pupil involvement in designing and delivering interventions, cited in the literature as a fundamental component to ensuring sustainability[26].

The most significant barrier to implementation identified by participants was that of curriculum pressure, as cited in recent research on TDM[33]. Headteachers and teachers discussed that the intense focus on academic targets and a curriculum tailored primarily to literacy and numeracy acted as a barrier to finding the time to implement TDM. In addition, academic expectations by parents exacerbated this problem, with parents questioning schools' allocation of time to physical activity over curriculum activities. Schools relied on trial and error in finding a time that fitted within the school structure and curriculum, requiring flexibility from teachers. The curriculum as a barrier to intervention implementation is well documented[26]. Research has suggested that until schools are assessed on health and wellbeing, interventions such as TDM will not be prioritised within the curriculum[27]. However within Wales the curriculum is currently undergoing a reform, with health and wellbeing constituting one sixth of the proposed new curriculum[52]. This reform creates potential for a shift in priority towards school-based programmes focussed on health outcomes such as TDM. It is therefore essential to provide schools with an evidence base on the effective implementation of TDM given its widespread political and media support that may pressure schools into uptake.

One method in overcoming the impact on curriculum time was suggested by teachers in this study who linked TDM with curriculum topics. Indeed, interventions that are integrated within the curriculum have been advocated for by teachers[27]. However, the suggestion that Daily Mile could possibly be being used as a replacement to PE is concerning and has been cited in another qualitative study of TDM[33]. Arguably, aside from physical activity, the wider aims and objectives of the PE curriculum such as play, motor skill development and physical literacy[28] are unlikely to be achieved through TDM alone. In addition, current provision of PE falls below national requirements of 120 minutes per week[29], and schools and pupils may benefit from encouraging regular quality PE provision rather than replacing with interventions. Findings from this study also highlight the importance of improving the quality of PE provision in primary schools so teachers feel confident in delivering PE as its own entity in addition to running programmes.

In this study, pupils currently not offered an afternoon playtime in school were positive about TDM providing a break from lessons. Research has highlighted the positive effect of active breaks on children's cognitive function and academic achievement[30–32]. The playground environment is considered a complimentary setting in promoting physical activity and play through unstructured activity and social interaction[29]. It is therefore unsurprising that pupils from another school expressed frustration about TDM replacing their afternoon playtime. However, recent research into the implementation of TDM[32] suggests that this caused less disruption to the school day and given the impact on learning time cited in this and other studies[33], headteachers may feel they have no choice. Play is an essential element of child development and the concerns over replacing playtime raised by pupils in this study demonstrate conflict between implementing TDM, curriculum pressure and the wider benefits to children's physical and social development offered throughout the school day.

There was conflict from schools regarding the competitive aspects of TDM, with a 'thrive or disengage' attitude. Some schools felt that it was important to deliver TDM as a non-competitive activity in line with the original principles. However, this conflicted with other teachers' perceptions of how to engage pupils who believed that the non-competitive element disengaged some pupils and rather, they thrived on friendly competition. In contrast, pupils were concerned about finishing last, highlighting the importance of delivering TDM as a continuous 15 minute activity, rather than the completion of a literal mile as supported by previous research[18]. Pupils setting personal goals and observing their progress encouraged participation and teachers suggested the use of rewards in increasing pupils' motivation. These

methods are supported by theoretical approaches to the promotion of PA utilising Bandura's social cognitive theory which models self-monitoring, goal setting and rewards[40].

Headteachers and teachers felt that TDM was more suited to the younger ages of primary school in which the curriculum is delivered through play[53], and observed that teachers of this age group were more engaged. In addition, the notion of teachers acting as role models through modelling behaviour and verbal encouragement was discussed by pupils and is supported in the literature[54]. However, a lack of teacher authority was highlighted by pupils as a concern in relation to pupils 'cheating' and not participating fully. Research has demonstrated the importance of involving school staff in the development of interventions to facilitate intervention ownership, autonomy and sustainability[29,55]. Teachers are agents of change in interventions, and teacher support and buy-in has been identified as a critical factor to implementation success [56,57]. Furthermore, teacher participation may elicit wider benefits such as improved teacher-pupil rapport, as identified in the literature[33]. The inclusion of teachers has also been advocated in a 'how to guide' developed by the University of Stirling, in which teacher participation and informal communication with pupils is encouraged[25].

This is further supported by findings from this study in which a whole-school approach to TDM was advocated, supported by pupils, teachers, leadership, parents and the wider community. Teachers also raised the challenge of a lack of external support by the Local Authority Sport and Health teams, although this was not an agreed responsibility prior to implementation. A 'Daily Mile Advisor' was suggested by one school who felt over-burdened with their role as coordinator. Previous research has indicated that schools have experienced initiative overload and a lack of collaboration between school-based programmes and the wider health field[55].

In addition, although one of the benefits cited by TDM is the lack of clothing or equipment required[25], this posed a challenge to schools. Issues of hygiene were discussed and a lack of appropriate footwear prevented participation. Weather conditions were also highlighted as causing concern in relation to clothing and safety and parental concerns.

Overall, views on the effect of TDM on pupils' behaviour and concentration were mixed. Some pupils felt their ability to concentrate in lessons and attitude to learning improved following participation in TDM. These immediate effects are supported by other qualitative research exploring implementation of TDM[32,33]. Research has also demonstrated a positive association between physical activity and cognition[58]. However, the suggestion that pupils were over-excited and behaviour was disrupted on return to the classroom is a concerning finding as this contradicts the benefits publicised by TDM such as improved behaviour and fuels the barrier of impact on learning time. In addition, pupils voiced that they felt tired and lacked energy which could account for teachers' perceptions of improved concentration and learning.

Pupils and teachers discussed the positive effect of TDM on pupils' physical, mental and social health and wellbeing. Pupils noted improvements in attitudes to PA, enhanced feelings of wellbeing and reduced feelings of stress. In addition, a number of social benefits were reported including displays of team-work and cooperation. In particular, TDM offered pupils the opportunity for social interaction with peers. The literature highlights mixed findings on the effect of school-based programmes on children's wellbeing due to the complex, multi-dimensional concept of wellbeing and inconsistencies in methodologies and measurement [59]. The general consensus however, is that school-based programmes contribute positively to health and wellbeing. Teachers also noted perceived improvements in pupils' CRF. This qualitative finding is supported by the exploratory analysis of the secondary aim conducted in this study which suggests that the CRF of children from south Wales increased between baseline and follow up following participation in TDM. An equal increase in the number of shuttles

run by the deprived and non-deprived groups was observed, however, the wide confidence intervals present within this data demonstrate the variability of changes in children's CRF. Adjusting for age and gender, there was no significant difference between children of high and low socio-economic groups. These results exhibit large standard deviation and therefore, strong conclusions on the association between TDM and children's CRF cannot be made.

In this sample of children from south Wales, UK, the deprived group performed a lower number of shuttles at both time points and thus, displayed a lower proportion of children classified as fit. The social gradient of physical activity, CRF and deprivation is demonstrated in the literature, with a larger proportion of children from a higher socio-economic status classified as fit compared to those from a lower socio-economic status[60]. Reducing inequalities in health and narrowing the deprivation gap in children's CRF is a public health priority, given the wide range of health benefits of regular physical activity[1,3]. Indeed, there is widespread recognition that universal school-based programmes that engage children from a range of socioeconomic backgrounds are effective in improving pupil health and wellbeing[13]. With the rapid uptake of TDM across schools in Wales and globally, it is important to examine whether its intended outcome of increasing children's PA and CRF are both valid and universal. In addition, previous research into school-based running programmes has identified unintended outcomes such as increased sedentary time outside of school compensating for the increased school PA[30]. However, no unintended outcomes were reported in this study. As implementation and adherence was not directly measured, in addition to limitations with the study design, it is difficult to draw conclusions on the overall effect of TDM on children's CRF. Furthermore, the large standard deviation and wide confidence intervals within this data suggest the need for future quantitative research to include larger samples in order to further examine and understand the impact on children's CRF.

## Strengths and limitations

This is the first mixed-methods study exploring the varying implementation and associated experience of TDM from a whole-school perspective and examining the association of TDM on children's CRF. Through incorporating headteachers', teachers' and pupils' views, this study provides important insights and recommendations for schools that contribute to the effective implementation of TDM in the future. With such widespread global adoption and expansion of TDM, this research is invaluable. However, a number of limitations are present in this study and it is important to consider these when attempting to draw conclusions from the findings.

School-based research poses a number of challenges in relation to the recruitment of schools. In this research study, a natural experimental approach was chosen due to the widespread adoption of TDM as a result of political and media support. However, the lack of a control group creates challenges in concluding the direct effect of TDM on children's CRF. Recruitment for this research study was conducted through convenience sampling, in which schools chose to begin implementing TDM at different time-points to coincide with school terms. This convenience sampling method could elicit selection bias as schools that volunteered to participate in the research study are likely to have a greater interest and investment in TDM with the potential to generate more positive feedback on implementation. In this research study, data collection was completed in two data collection phases due to schools choosing to implement TDM at different time-points. Research has identified the effect of seasons on PA, with lower levels of MVPA exhibited in autumn and winter[61]. Statistical analyses did not adjust for season and this should be taken into account when interpreting findings. However, the strength of this from a qualitative perspective is that experiences are captured

throughout the academic year. Furthermore, schools had varying exposure to TDM and a dose-response relationship may impact on CRF. Finally, it must be considered that changes in CRF could be due to a number of other factors aside from TDM such as growth and maturation[62] and improvement in 20m SRT participation.

The implementation and fidelity of TDM was not directed measured in this study, although anecdotal differences are reported through qualitative findings. All schools were invited to participate in pupil focus groups and interviews with teachers and headteachers. However, not all schools participated in all three qualitative measures and findings represent those that chose to participate. This may impact the transferability of results. Focus group sampling was achieved through teachers selecting consented pupils fulfilling a criteria of mixed gender and physical activity abilities. Although teachers were reminded of the importance of selecting a variety of pupils with a range of abilities in order to capture a variety of experiences, there is potential that this method could cause bias in results through preferential selection. Future research would benefit from an in-depth process evaluation of a larger sample of schools. Triangulation of findings could help highlight of the strength and weaknesses of implementation factors on outcomes.

## Conclusions

Findings from this study have identified a range of barriers and facilitators to implementing and sustaining TDM from a whole-school perspective. The schools in this study varied the implementation and this is reflected in the differing perspectives and experiences of participants. Ultimately, the implementation of TDM affected the pupils' enjoyment, participation, experience and potential for sustainability. For future effective implementation and longer-term sustainability of TDM the findings from this study recommend;

- Flexible, adaptable implementation incorporating pupil feedback

- Delivered during curriculum time (excluding Physical Education) or as an afternoon play-time (not replacing current play provision)

- Encouraging individual competition through personal goal setting (incorporated into curriculum work)

- Active involvement and participation of teachers and staff

- Whole-school and wider community support, engaging with parents, community stakeholders and local sporting role models

## Supporting information

**S1 Appendix.** Schematic diagram of data collection time points (baseline and follow up)–Schools A-F.
(DOCX)

**S2 Appendix. Interview and focus group participation.**
(DOCX)

**S3 Appendix. Focus group and interview topic guides.**
(DOCX)

**S4 Appendix. Themes and sub-themes.**
(DOCX)

**S5 Appendix.** Descriptive characteristics of shuttles and CRF by school (A-F). (DOCX)

## Acknowledgments

The authors would like to thank all participating schools, headteachers, teachers and pupils that took part in this study. Infrastructure support was received by the National Centre for Population Health and Wellbeing Research.

## Author Contributions

**Conceptualization:** Emily Marchant, Sinead Brophy.

**Data curation:** Emily Marchant, Charlotte Todd.

**Formal analysis:** Emily Marchant, Charlotte Todd.

**Investigation:** Emily Marchant.

**Methodology:** Emily Marchant, Charlotte Todd, Gareth Stratton, Sinead Brophy.

**Project administration:** Emily Marchant.

**Resources:** Emily Marchant.

**Supervision:** Gareth Stratton, Sinead Brophy.

**Writing – original draft:** Emily Marchant, Charlotte Todd.

**Writing – review & editing:** Emily Marchant, Charlotte Todd, Gareth Stratton, Sinead Brophy.

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
