## [Decision Letter · Decision Letter 0]

26 Sep 2019

PONE-D-19-21022

The Daily Mile can equally improve cardiorespiratory fitness in deprived and non-deprived children. Whole-school recommendations for implementation and sustainability: a mixed-methods study.

PLOS ONE

Dear Miss Marchant,

Thank you for submitting your manuscript to PLOS ONE. After careful consideration, we feel that it has merit but does not fully meet PLOS ONE’s publication criteria as it currently stands. Therefore, we invite you to submit a revised version of the manuscript that addresses the points raised during the review process.

ACADEMIC EDITOR: You will see that both reviewers are agreed that this manuscript could potentially make an important contribution to the literature. However, they have raised some concerns about the reporting of the methods and findings, and have made a number suggestions on how the transparency of the study could be improved. One reviewer has suggested reframing the paper to focus only on the qualitative components. If you decide not to implement this recommendation, please provide a robust rebuttal in your response, which we look forward to receiving.

We would appreciate receiving your revised manuscript by Nov 10 2019 11:59PM. To enhance the reproducibility of your results, we recommend that if applicable you deposit your laboratory protocols in protocols.io, where a protocol can be assigned its own identifier (DOI) such that it can be cited independently in the future. For instructions see: http://journals.plos.org/plosone/s/submission-guidelines#loc-laboratory-protocols

We look forward to receiving your revised manuscript.

Kind regards,

Shelina Visram, PhD, MPH, BA

Academic Editor

PLOS ONE

Journal Requirements:

Additional Editor Comments (if provided):

Reviewers' comments:

Reviewer's Responses to Questions

**Comments to the Author**

1. Is the manuscript technically sound, and do the data support the conclusions?

Reviewer #1: Partly

Reviewer #2: Partly

2. Has the statistical analysis been performed appropriately and rigorously? 

Reviewer #1: I Don't Know

Reviewer #2: I Don't Know

3. Have the authors made all data underlying the findings in their manuscript fully available?

Reviewer #1: Yes

Reviewer #2: No

4. Is the manuscript presented in an intelligible fashion and written in standard English?

Reviewer #1: Yes

Reviewer #2: Yes

5. Review Comments to the Author

Reviewer #1: Thank you for the opportunity to review his manuscript. The manuscript reports on mixed-methods pilot study were to compare (1) the effect of TDM on the cardiorespiratory fitness (CRF) of children in high and low socio-economic groups and (2) explore whether children’s experiences of TDM was related to implementation. The exponential growth of The Daily Mile and its reach among schools suggests that it has significant potential as a public health programme however, given the very small evidence base surrounding school-based running programmes and limited information on the impact that this has had, I am supportive of the concept of the study and feel that such a study would make a contribution to the evidence base. However, the study addresses two very different topics related to TDM and does not attempt to align the two. In addition, it suffers from some methodological weaknesses and limitations e.g. small sample size, large amount of imputed data, differing levels and short duration of implementation and no direct measure of implementation or fidelity. Much more detail is also needed throughout the materials and methods section in particular. I therefore recommend that this paper is accepted subject to major revisions and I would be happy to review a revised draft. Further details are provided below.

Title

The authors state that the current study is a pilot study (line 29), consider revising the title to ensure that this is reflected.

Introduction

• As PlosOne targets an international audience, it may be useful to add one or two sentences to outline the importance of promoting PA in children and current prevalence rates / issues in school-based PA in Wales/UK. Similarly, is there any information available to indicate how many schools are adopting TDM in Wales?

Materials and methods

• Can you please include any information on theory used to plan your study, if any?

• To ensure methodological coherence the authors should explicitly state how their philosophical worldviews informed their study design

Study design

• It might be useful for the reader to consider a figure or schematic to represent the differing data collection time points between the schools, this is just a suggestion.

• Did the authors consider examining whether fitness level of pupils differed between schools depending on level of implementation of TDM? This would be interesting to examine and would help to align the two themes within the paper – currently this they are very separate and a little out of place.

Participants and setting

• Information regarding the percentage of pupils eligible for free school meals is presented in line 134 but would it be possible to add some more information on the demographics and characteristics of the schools (e.g. size, location etc. In addition, do you know/ have any information on whether any of the schools had experience of implementing a school-based running programme previously? This is really important contextual information which is likely to influence the implementation of TDM.

• Much more information is needed on the recruitment of schools to the study and recruitment of participants to the different elements. Did any of the schools contacted through HAPPEN decline the offer to participate in the study? If so were the characteristics of the six schools recruited to participate similar or different to those that declined? Such recruitment information is important to include. How were the pupils recruited to participate in the shuttle run test, was this all Yr 5/6 pupils in all schools or a sample? Similarly, how were pupils and staff recruited for the interviews and focus groups?

Qualitative measures

• More detail on the development of the focus group and interview guides is needed e.g. were they theory informed? Literature informed?

• Please provide some information on the duration of the focus groups and interviews.

Quantitative analysis

• As per previous comment, please be explicit in providing information on participant numbers – there is a large volume of imputed data, please comment on what basis the authors assume that data is missing at random? It is likely that some children may have been absent because of awareness of the forthcoming testing and were not motivated to participate.

Qualitative analysis

• Much more detail on the analysis is needed; a description of each step and what each step looked like in the context of this research is necessary. Was a coding manual developed prior to their analysis to guide those coding their data? As such, were themes identified prior to collecting the data or following review of the transcripts? Assuming both the interviews and focus groups were transcribed verbatim (reference is only made to the interviews) was this into Microsoft Word? Was computer-assisted qualitative data analysis software such as NVIVO used? How was rigour and trustworthiness of the findings maintained?

Results

• The results would benefit from the reporting of the descriptive characteristics of the total sample, as well as for those who participated in the CFR assessments and the focus groups. At the moment, it is not clear where the missing data came from, nor which pupils were included in the focus groups or which schools they came from. How generalizable are these responses?

Results – qualitative

• One of the biggest debates at the moment is around defining a schools’ participation in TDM. Did you explicitly use the original principles as a way of distinguishing schools? It may be useful to include (as supplementary information) or refer to these within the text given the overlap with some of the themes.

• The quotations should represent the range/scope of the responses, including minority response. The majority of the teacher quotes come from schools A and B and no pupil quotes are provided from schools A, B or C – however, it is not possible to know if this is because teachers and pupils from these schools did not participate in the focus groups or if they did not say anything in support of or different to these themes because it hasn’t been reported in the methods. Similarly, are the authors able to give an indication as to how many pupils/teachers said or agreed with the statements e.g. a few, some, many, all. At the moment each statement sounds as if these themes/findings are universal across all participants.

Discussion

• There is very little discussion of the findings related to CRF in the context of other research e.g. are the authors able to comment on how the levels of fitness described in the current study compare to other studies on this population?

• There is a lot of repetition from the results section, consider making more concise. Are the authors able to tie in and align findings and discussion on CFR with implementation at all, at the moment it appears quite disjointed.

• You have outlined a number of barriers and facilitators to implementation of TDM in school. It would be worthwhile also consulting with systematic reviews of schools based physical activity programs to expand this list and ensure it is consistent with recent literature. You may find the following references helpful when thinking about running programmes specifically:

Malden S, Doi L. The Daily Mile: teachers’ perspectives of the barriers and facilitators to the delivery of a school-based physical activity intervention. BMJ Open 2019;9:e027169. doi:10.1136/ bmjopen-2018-027169

Chalkley AE, Routen AC, Harris JP, Cale LA, Gorely T., Sherar LB. (2018) A retrospective qualitative evaluation of barriers and facilitators to the implementation of a school-based running programme. BMC Public Health, 18:1189

• Are the authors able to suggest ways future research can confirm findings and or build on what has been learnt? The authors state that the current study is a pilot study (line 29), which implies that the results will inform a full-trial. I would question what scope there is for a full trial of the intervention if it has already been implemented. It could be that it is being phased out region by region, in which case it would be possible to do a pilot and then a larger scale evaluation in a new area, but it should be clear in the paper.

• A number of practical recommendations to support the implementation of running programmes has been provided and the authors may find it useful to refer to these in the discussion. See Chalkley et al 2018 (as above) and https://4715cv8pjis4a4bp224xekg1-wpengine.netdna-ssl.com/wp-content/uploads/2018/10/How-To-Why-To-Guide.pdf

Strengths and limitations

• The natural experiment design has the potential to introduce bias. Some acknowledgements of this bias as a limitation would be worthwhile as well as the design’s ability to isolate the effect on CRF and the numerous other variables which could influence it e.g. maturation, seasonality, physical activity levels etc

• Convenience sampling schools is likely to introduce some bias. Acknowledgement of this and what would be deemed as best practice is required.

• In addition, if I have understood it right, the differing length of exposure to TDM should be acknowledged i.e. School A was implementing for 6 months, schools C-F for 5 months and School B for 2 months, and implications of this on the quantitative and qualitative data collected e.g. limits ability to establish how such programmes are maintained in schools.

Conclusion

• I do not feel as if the results are supportive of the conclusion in relation to CRF due to the reasons outlined previously.

References

• Please check your referencing system. There are missing references e.g. no 40 (line 724) as well as inappropriate references provided.

Supplementary files

• I would suggest reordering the guides in the supplementary files so that baseline interview/focus group guide is presented first.

Reviewer #2: The paper describes the apparent TDM effects on cardiorespiratory fitness in 9-11 year old children from South Wales. Personally, I have concerns about the way this outcome has been interpreted, which I discuss more below. For me, the far more important part of the paper from an implementation and scientific perspective is the qualitative work, which sheds light on crucial aspects of implementing and evaluating the TDM from the teacher and child perspective. Due to my concerns about the CRF element (listed below), I would like to suggest the authors consider flipping the focus of the manuscript to the qualitative aspect, with far less emphasis placed on the CRF data presented. My reasons for this suggestion and other specific comments can be found below.

Abstract

“Both groups demonstrated equal increases in shuttles between baseline and follow-up (deprived: 4.7 ± 13.4, nondeprived: 4.8 ± 16.0). There was no significant difference in the increase in shuttles run for deprived compared to non-deprived children adjusted for age and gender. This finding suggests that TDM is an effective universal tool in improving children’s CRF.”

Given the relatively small change in 20mSRT performance (less than half a level on the shuttle run test) and the large SD associated with the change in both groups (highlighted above), I do not think the statement the ‘TDM is an effective universal tool in improving children’s CRF’ is supported. I suggest softening greatly, and also acknowledging the sample (children from South Wales) within any statement.

Introduction

1. If the purpose of the introduction is to provide rationale for the need to increase physical activity levels, then the first few paragraphs of the introduction are very strong. The focus of the paper however seems to be on cardiorespiratory fitness, which (as I’m sure the authors know) is distinctly different from physical activity. The fact that cardiorespiratory fitness is not mentioned until the aims of the study are listed concerns me, as no rationale for examining this outcome has really been provided. The introduction reads as though it is physical activity that is going to be measured, not cardiorespiratory fitness. This should be addressed in the revision process and I suggest the authors consider directing the focus of this manuscript away from the CRF element.

2. The paper cited in reference 13 (e.g. Chesham RA, Booth JN, Sweeney EL, Ryde GC, Gorely T. The Daily Mile makes primary school children more active , less sedentary and improves their fitness and body composition : a quasi-experimental pilot study. BMC Med. 2018) has been heavily critiqued and the findings discredited by several sources due to major limitations in the study design. Please consider and discuss these critiques in a lot more detail for transparency.

3. Can the authors please provide some more context as to why it is important to look at deprived and non-deprived cohorts separately? Please contextualise this to Wales in particular?

4. Please strongly consider changing the focus of the manuscript to the second aim (e.g. explore whether children’s experiences of The Daily Mile was related to implementation) and mention CRF as an exploratory secondary outcome only. The qualitative element is so strong and I feel the CRF element detracts from this excellent work and weakens the paper overall.

Material and methods

1. Personally, I do not think the study design utilised here allows for such strong conclusions on CRF to be made. This should be recognised in study design section and is the main reason I do not think CRF should be the focus of this manuscript. While conducting a natural experiment is pragmatic and should be commended, limitations within the study design does not allow confidence in the conclusions drawn from this study. Firstly, as of course there is no control group, it is impossible to ascertain whether the pupils involved in study increased their CRF due to TDM, or because of some other factors (e.g. growth and maturation, measurement error within the test, seasonal effects, or pure coincidence). This has to be addressed throughout the manuscript and conclusions adapted throughout. I also think it should be made clearer why post-intervention measures were collected at three months in some schools and six months in others. In terms of dose, this suggests some schools had double the dose of the intervention than others, which likely will have impacted the findings. Again, I am aware the pragmatic nature of the study design may have dictated this, but this is another reason for weakening the statements on CRF improvement and changing the manuscript focus towards implementation instead.

2. Qualitative measures: Can you please provide information on how pupils were recruited into the focus group part of the study? Was consent for this element separate from the 20mSRT? Can you also indicate whether all pupils involved in the 20mSRT had the opportunity to take part in the focus groups, or whether a sub-sample were recruited? If the latter, how was this achieved?

3. Qualitative analysis: Please provide more detail in the steps conducted for the qualitative synthesis of the data.

Results

“However, the number of children who were classed as fit in the deprived group increased by 14% compared to an 8% increase in the non-deprived children”.

I am not sure how relevant this statement is, as surely it is obvious more children in the deprived group were classed as fit post-intervention, due to the differences between the groups at baseline? E.g. more kids in the non-deprived group were already in the ‘fit’ group as baseline, therefore could not progress to a ‘higher’ group as there was nowhere to go?

Due to the issues I have with the study design, I cannot comment much further on aspects of this analysis as I am not sure how relevant it is given the limitations of the study design. I would however like the authors to comment on the whether a change in ~5 shuttles over a 3-6 month programme is actually meaningful. I would also like the authors to comment on the large SD associated with the change in CRF, and the impact this has on the interpretation of the findings.

The authors should be highly commended on the qualitative aspects of this study. The data is rich and highly useful for future implementation and as a potential means of exploring the fidelity of TDM. Unfortunately, much of this excellent work gets lost given the (in my opinion) incorrect primary focus on CRF in the manuscript. Given the concerns I have with the CRF element of this study, I do not think it is appropriate to review the associated sections in the discussion at this time. I will therefore reserve my comments on the qualitative discussion section until after the revision process.

6. PLOS authors have the option to publish the peer review history of their article (what does this mean?). If published, this will include your full peer review and any attached files.

Reviewer #1: No

Reviewer #2: Yes: Kathryn Weston

---

## [Author Response · Author response to Decision Letter 0]

4 Nov 2019

Editor

One reviewer has suggested reframing the paper to focus only on the qualitative components. If you decide not to implement this recommendation, please provide a robust rebuttal in your response, which we look forward to receiving.

Thank you for inviting me to submit a revised version of the manuscript. I agree with the reviewer’s suggestion to reframe the focus of the paper and focus on the qualitative component and believe this significantly strengthens the paper. As a result, I have focussed the primary aim of the study on exploring participants’ experiences of The Daily Mile (TDM) to gain an understanding of how implementation is related to experience. I have included the quantitative element as a secondary aim to examine the association between TDM and children’s cardiorespiratory fitness and to compare this association between children in high and low socio-economic groups. This focus throughout the manuscript on the qualitative findings concludes with a set of recommendations for the future effective implementation of TDM. Given the rapid, widespread adoption of TDM and the public health and political support for implementation, these findings support schools in their implementation and sustainability from a whole-school perspective. Although a secondary aim, the quantitative element adds to the limited evidence on TDM and is discussed in relation to supporting the qualitative finding of perceived improvements in children’s CRF. 

Reviewer #1

As PlosOne targets an international audience, it may be useful to add one or two sentences to outline the importance of promoting PA in children and current prevalence rates / issues in school-based PA in Wales/UK. Similarly, is there any information available to indicate how many schools are adopting TDM in Wales

Response

Thank you for this suggestion. I agree that providing more context on the importance of promoting PA and the prevalence rates in Wales is important. To address this, I have included recent survey data on children’s PA levels from the Active Healthy Kids Wales Report Card (2018). Active Healthy Kids is a global alliance that allows comparison across all 49 countries that take part. I have also updated the number of schools participating in TDM in Wales and globally. 

Change

Page 3: Furthermore, survey level data from latest Active Healthy Kids Wales Report Card within Wales, United Kingdom suggests that just 34% of children aged 3-17 years are meeting these guidelines [9]. In response to this data, the expert research group concluded the need to strengthen efforts in creating opportunities that increase children’s PA. This group also highlight the gap in nationally representative data [9].

Page 5: The intervention’s simple design and replicability has resulted in rapid uptake and is now being delivered in over 480 schools in Wales, and over 10000 schools worldwide[22]. 

Can you please include any information on theory used to plan your study, if any?

Response

This work forms a case study design where we examined six primary schools and their experiences of implementing the Daily Mile. The purpose of this research study was not to develop theory, but to explore the experience of participants in relation to implementation of TDM in order to inform future practice and sustainability. 

Change

Page 8: The purpose of this study was to understand the experiences of participants in relation to implementation of TDM to inform future practice and sustainability rather than to develop new theory.

To ensure methodological coherence the authors should explicitly state how their philosophical worldviews informed their study design

Response

Thank you for this suggestion. I have included this within the Materials and Methods section. 

Change

Page 11: The qualitative component of this research study adopted an interpretive approach through thematic analysis in order to gain an understanding of participants’ experiences of implementing TDM.

It might be useful for the reader to consider a figure or schematic to represent the differing data collection time points between the schools, this is just a suggestion.

Response

Thank you for raising this point. I appreciate that the different phases of data collection may be confusing for the reader to interpret and agree that a schematic representation of the different phases and data collection time points would provide clarity. 

Change

Page 8: A schematic diagram representing data collection periods across schools A-F is provided for clarity in the supporting information (S1 Appendix). 

Did the authors consider examining whether fitness level of pupils differed between schools depending on level of implementation of TDM? This would be interesting to examine and would help to align the two themes within the paper – currently this they are very separate and a little out of place.

Response

This is an interesting suggestion and thank you for raising this. The level of implementation was not directly measured in this study. The implementation style was reported by participants and emerged from the qualitative findings. I have now stated this explicitly in the methodology and suggested that this would be useful to measure in future research. 

In response to your suggestion that the two themes need to be aligned, I believe the restructured focus on the qualitative findings and the set of recommendations is an important addition to the literature and practical guidance for schools for the future delivery of TDM. Given that participants reported perceived improvements in children’s fitness, I have structured the quantitative findings to support this qualitative finding. Furthermore, there remains a limited evidence base on quantitative findings of TDM and I believe it remains important to report on within this manuscripts. Regarding school-based programmes such as TDM, schools often want to know “does it work” and “how should it be delivered”. The authors feel both elements of this paper are an important contribution to the literature. 

Change

Page 8: Implementation level of TDM was not directly measured in this study but rather, emerged anecdotally through qualitative analysis. 

Page 33: However, implementation was not directly measured and future research into TDM would benefit from examining the strength of outcomes in relation to implementation level and style.

Information regarding the percentage of pupils eligible for free school meals is presented in line 134 but would it be possible to add some more information on the demographics and characteristics of the schools (e.g. size, location etc)

Response

Thank you for this suggestion. I agree that additional school demographic information would be useful for the reader and I have addressed this with as much detail as possible whilst maintaining anonymity of schools and participants. However, I am unable to include exact school size and location as this is identifiable information. 

Change

Page 9: The school size ranged from 175 to 275 pupils. 

In addition, do you know/ have any information on whether any of the schools had experience of implementing a school-based running programme previously? This is really important contextual information which is likely to influence the implementation of TDM.

Response

This is an interesting point and I agree that a school’s previous experience of implementing similar programmes is important contextual information for the reader. 

Change

Page 9: Schools had minimal experience of implementing previous whole-school running programmes. 

Much more information is needed on the recruitment of schools to the study and recruitment of participants to the different elements. Did any of the schools contacted through HAPPEN decline the offer to participate in the study? If so were the characteristics of the six schools recruited to participate similar or different to those that declined? Such recruitment information is important to include

Response

Thank you for this suggestion. I agree that further clarity is needed on the recruitment process. The authors cannot comment on schools declining to participate as initial recruitment was facilitated through this Sports Development officer. We also cannot comment on whether schools that didn’t participate in the research chose to decline or were already implementing TDM. This remains a challenge with school-based programmes that expand so rapidly. However, I have provided further information regarding recruitment that is outlined below. 

Change

Page 8-9: The initial school recruitment process was facilitated through an Active Young People (AYP) Officer from the Local Authority’s Sports Development team through an existing partnership with HAPPEN. The AYP officer had established links with all primary schools in their cluster area within the Local Authority and emailed these schools with an expression of interest in implementing TDM. Six primary schools (Schools A-F) responded and were subsequently contacted through HAPPEN via email regarding their intention to implement TDM. Recruited schools were then contacted via a telephone conversation with the headteacher. 

How were the pupils recruited to participate in the shuttle run test, was this all Yr 5/6 pupils in all schools or a sample? Similarly, how were pupils and staff recruited for the interviews and focus groups?

Response

Thank you for raising this point. I agree that more information is needed regarding recruitment to the separate components. 

Change

Page 9: All pupils from years 5&6 from schools A-F were invited to participate in both the qualitative and quantitative measures. Pupils had the option to consent to participate in one or both measures in consent forms. Headteachers and all teachers from years 5&6 from the six schools were invited to participate in the qualitative measure. Headteachers and all teachers from years 5&6 from the six schools were invited to participate in the qualitative measure.

Page 10: Each focus group was conducted by year group and consisted of between six and eight pupils [40] aged 9-11 years of mixed physical activity ability and gender. Class teachers were provided with a list of consented pupils and selected pupils fulfilling this criteria.

More detail on the development of the focus group and interview guides is needed e.g. were they theory informed? Literature informed?

Response

Thank you for bringing this to my attention. I agree that further detail regarding the development of topic guides would be useful for the reader. Please find details below. 

Change

Page 10: All interviews and focus groups followed a semi-structured topic guide, initially developed by EM and CT and reviewed by SB to address the qualitative research aims. In order to explore participants’ experiences of TDM, it is important to consider the barriers, facilitators and factors affecting sustainability. These factors are consistently included in other research evaluating school-based interventions, and therefore framed the topics guides for this study [33]. The use of semi-structured topic guides facilitated a deeper exploration of subjects and allowed topics to form naturally during the interview process [41]. These topic guides were not piloted prior to data collection but were based on previous school-based programme research through HAPPEN[42]. 

Please provide some information on the duration of the focus groups and interviews.

Response

Thank you for raising this point, I agree that this is important information.

Change

Page 10-11: The duration of interviews ranged between 5 and 21 minutes and focus groups between 23 and 48 minutes. 

As per previous comment, please be explicit in providing information on participant numbers – there is a large volume of imputed data, please comment on what basis the authors assume that data is missing at random? It is likely that some children may have been absent because of awareness of the forthcoming testing and were not motivated to participate.

Response

Thank you for raising this point. I have updated the results tables and included an ‘Overall’ column with participant numbers to provide more transparency on imputed and non-imputed data. The authors assumed data to be missing at random as there was no significant difference of baseline shuttles between groups (missing at follow up, present at follow up). Further to the changes below, there were more children present at the follow-up testing and the authors feel this is an indication of motivation to participate.

Change

Page 13: Data were assumed to be missing at random (e.g. probability of being missing does not depend on the missing value) on the basis that there was no significant difference of baseline shuttles between groups (missing at follow up, present at follow up). 

Page 32: Updated table 2. Descriptive characteristics (overall, deprived, non-deprived)

Much more detail on the analysis is needed; a description of each step and what each step looked like in the context of this research is necessary. Was a coding manual developed prior to their analysis to guide those coding their data? As such, were themes identified prior to collecting the data or following review of the transcripts?

Response

Thank you for bringing this to my attention. I agree that further clarity is needed on the qualitative analysis process. I have further described the steps that were taken in this research study, outlined by Burnard (1991). 

Change

Page 12: The process of analysing the interview and focus group data followed the steps outlined by Burnard (1991) [48]. To begin, each transcript was independently read several times by two researchers (EM and CT) to facilitate immersion in the data. The researchers (EM and CT) then followed an independent open coding process to allow participants’ views to be summarized by assigning words or phrases to quotes or paragraphs. This initial list of freely generated categories following review of the transcripts aimed to encapsulate interviewees’ responses and were subsequently grouped according to the overarching theme. Through this process, broader categories were combined to produce one higher-order heading that captured the overall meaning of responses. This process was repeated whereby similar categories were synthesised to produce a final list of themes and sub-themes. Both researchers (EM and CT) compared their lists of themes and sub-themes to ensure accuracy and consistency. If there was a discrepancy or disagreement in coding, a third researcher (SB) adjudicated. This method enhances the validity of categories assigned and attempts to reduce researcher bias[50]. The written notes taken on the day of the interview or focus group were compared with these topics to ensure an accurate account of participants’ responses. Following this, the two researchers worked together through an extensive process to discuss codes and categorise them under final themes and sub-themes (S4 Appendix). The lead researcher (EM) then manually worked through each transcript and coded the responses according to the final list of themes and sub-themes. All responses grouped by themes and sub-themes were compiled to a master copy document that was used for reference to write up the findings. 

Assuming both the interviews and focus groups were transcribed verbatim (reference is only made to the interviews) was this into Microsoft Word?

Response

Thank you for bringing this oversight to my attention. I have addressed this within the manuscript. 

Change

Page 11-12: All interviews and focus groups were digitally recorded and transcribed verbatim in Microsoft Word.

Was computer-assisted qualitative data analysis software such as NVIVO used?

Response

Thank you for raising this point. No computer-assisted qualitative data analysis software was used. I have updated the Qualitative Analysis to further explain the process undertaken. 

Change

Page 11-12: All interviews and focus groups were digitally recorded and transcribed verbatim in Microsoft Word. The process of analysing the interview and focus group data followed the steps outlined by Burnard (1991) [48]. To begin, each transcript was independently read several times by two researchers (EM and CT) to facilitate immersion in the data. The researchers (EM and CT) then followed an independent open coding process to allow participants’ views to be summarized by assigning words or phrases to quotes or paragraphs. This initial list of freely generated categories following review of the transcripts aimed to encapsulate interviewees’ responses and were subsequently grouped according to the overarching theme. Through this process, broader categories were combined to produce one higher-order heading that captured the overall meaning of responses. This process was repeated whereby similar categories were synthesised to produce a final list of themes and sub-themes. Both researchers (EM and CT) compared their lists of themes and sub-themes to ensure accuracy and consistency. If there was a discrepancy or disagreement in coding, a third researcher (SB) adjudicated. This method enhances the validity of categories assigned and attempts to reduce researcher bias[50]. The written notes taken on the day of the interview or focus group were compared with these topics to ensure an accurate account of participants’ responses. Following this, the two researchers worked together through an extensive process to discuss codes and categorise them under final themes and sub-themes (S4 Appendix). The lead researcher (EM) then manually worked through each transcript and coded the responses according to the final list of themes and sub-themes. All responses grouped by themes and sub-themes were compiled to a master copy document that was used for reference to write up the findings. 

How was rigour and trustworthiness of the findings maintained?

Response

This is a useful point to raise, thank you for bringing this to my attention. 

Change

Page 11: The lead researcher (EM) facilitated the interview process, whilst the other researcher provided technical support (digitally recording) and made field notes on key responses. At the start of each interview and focus group, researchers reminded the participants of the study aims, guidelines on anonymity and confidentiality and encouraged participants’ personal viewpoints. In order to achieve neutrality, researchers emphasised that they remained impartial and there were no right or wrong answers. In order to gain respondent validation, these notes were verbally summarised through member checking with interviewees at the end of each interview. To ensure trustworthiness, the researcher’s interpretation of responses were summarised for corrections, clarification or confirmation by participants[43,44].

Page 12: To begin, each transcript was independently read several times by two researchers (EM and CT) to facilitate immersion in the data.

Page 12: Both researchers (EM and CT) compared their lists of themes and sub-themes to ensure accuracy and consistency. If there was a discrepancy or disagreement in coding, a third researcher (SB) adjudicated. This method enhances the validity of categories assigned and attempts to reduce researcher bias [48]. The written notes taken on the day of the interview or focus group were compared with these topics to ensure an accurate account of participants’ responses.

The results would benefit from the reporting of the descriptive characteristics of the total sample, as well as for those who participated in the CRF assessments and the focus groups. 

Response

Thank you for raising this point. I agree that a further breakdown of descriptive characteristics would strengthen the results section. The authors are unable to provide a further breakdown of focus group participants as the final list of pupils was discarded to ensure anonymity.

Change

Page 31: Updated Table 1. Descriptive characteristics (participated in 20m SRT, total sample) 

Page 10: Class teachers were provided with a list of consented pupils and selected pupils fulfilling this criteria. This list was discarded following selection of pupils and a final list of pupils participating in focus groups was not recorded to ensure anonymity.

At the moment, it is not clear where the missing data came from, nor which pupils were included in the focus groups or which schools they came from. How generalizable are these responses?

Response

Thank you for raising this point. Please find details below addressing your queries regarding missing data and focus group participation. I have updated the tables within the results section with additional participant numbers and an ‘overall’ column. I cannot provide descriptive characteristics of which pupils participated in focus groups as the list of final participants was discarded by the teacher. However, I have included an additional table in the Qualitative Measures section which outlines the number of interviews and focus groups per school at each time point. This has also been acknowledged in the Strengths and Limitations section. 

Change

Page 10: A further breakdown of interviews and focus group participation by school can be found in the supporting information (S2 Appendix).

Page 10: Class teachers were provided with a list of consented pupils and selected pupils fulfilling this criteria. This list was discarded following selection of pupils and a final list of pupils participating in focus groups was not recorded to ensure anonymity.

Page 30: Table 1 presents the descriptive characteristics of those that participated in CRF tests and the total sample (including imputed data). A total of 229 pupils participated in the 20m SRT at baseline and 235 pupils at follow up. In total, 204 pupils completed the 20m SRT at both time points. The MICE imputation method utilising shuttles, age and deprivation accounted for an additional 34 pupils at baseline and 28 pupils at follow up.

Page 31: Updated Table 1. Descriptive characteristics (participated in 20m SRT, total sample). 

Page 32: Updated Table 2. Descriptive characteristics (overall, deprived, non-deprived). 

Page 40: All schools were invited to participate in pupil focus groups and interviews with teachers and headteachers. However, not all schools participated in all three qualitative measures and findings represent those that chose to participate. This may impact the transferability of results. 

One of the biggest debates at the moment is around defining a schools’ participation in TDM. Did you explicitly use the original principles as a way of distinguishing schools? It may be useful to include (as supplementary information) or refer to these within the text given the overlap with some of the themes

Response

Thank you for raising this point. I agree that there is a wide variation in what is considered gold standard in terms of a school’s participation in TDM. In this research study, there was no strict definition of a schools participation in TDM. Schools were committed to delivering TDM as a whole-school running programme and were aware of the original principles. 

The level of implementation was not directly. However, questions regarding delivery were explored in the interviews and focus groups to examine differences in implementation and associated experience. As the qualitative findings demonstrate, there was a great deal of variability in the delivery style of TDM. These findings are summarised through the discussion section in relation to the varying delivery themes and associated experience. This remains a challenge in school-based interventions. 

Change

Page 34: Conflict existed between schools on how TDM should be delivered, raising the issue of fidelity to the intervention.

Page 34: However from a research perspective, variation in delivery and fidelity to the original intervention design poses a number of challenges for evaluating school-based interventions.

Page 40: The implementation and fidelity of TDM was not directed measured in this study, although anecdotal differences are reported through qualitative findings. 

The quotations should represent the range/scope of the responses, including minority response. The majority of the teacher quotes come from schools A and B and no pupil quotes are provided from schools A, B or C – however, it is not possible to know if this is because teachers and pupils from these schools did not participate in the focus groups or if they did not say anything in support of or different to these themes because it hasn’t been reported in the methods.

Response

Thank you for raising this point. I have updated the supporting information (S2 Appendix) to indicate transcript numbers, participants, school and time point of focus groups and interviews. Schools A-C did not participate in pupil focus groups. However, all schools were recruited to participate in all measures. I have amended the Methods and Materials section to state this. In addition, I have acknowledged this within the Strengths and Limitations section. 

Change

Page 9: All pupils from years 5&6 from schools A-F were invited to participate in both the qualitative and quantitative measures. Pupils had the option to consent to participate in one or both measures in consent forms. Headteachers and all teachers from years 5&6 from the six schools were invited to participate in the qualitative measure. 

Page 10; A further breakdown of interviews and focus group participation by school can be found in the supporting information (S2 Appendix).

Page 40: All schools were invited to participate in pupil focus groups and interviews with teachers and headteachers. However, not all schools participated in all three qualitative measures and findings represent those that chose to participate. This may impact the transferability of results. 

Similarly, are the authors able to give an indication as to how many pupils/teachers said or agreed with the statements e.g. a few, some, many, all. At the moment each statement sounds as if these themes/findings are universal across all participants.

Response

Thank you also for raising the point of how many pupils/teachers said or agreed with statements. Please find details below where I have updated these statements within the manuscript where possible;

¬Change

Page 15: However, a consistent implementation method that maintained the original principles was associated with a lack of enthusiasm and engagement by many participants; 

Page 16: One of the main concerns for all headteachers and teachers was the curriculum pressure by educational inspectorates.

Page 16: Therefore, incorporating flexibility lead to a more positive experience and increased engagement from most pupils. 

Page 17: For many pupils, the replacement of their playtime was a negative factor associated with their experience

Page 17: The addition of an afternoon playtime to participate in TDM was supported all by pupils

Page 19: Many teachers also recognised the need for rewarding pupils to encourage their participation, stating that the wider benefits of participation were not valued by pupils

Page 21: In contrast, some pupils from other schools discussed the passive involvement of teachers and the negative effect this had on participation. For pupils, disengaged teachers resulted in disengaged pupils and rule breaking

Page 21: However, some schools discussed the conflict that existed between the engagement of teachers from different year groups;

Page 22: The varying level of support from staff, parents, stakeholders and the wider community were discussed by many participants

Page 24: Some teachers questioned the seasonal effects of TDM for pupils, speculating that pupils’ enthusiasm was dependent on the weather;

Page 24: However, despite summer being seen as an ideal term to implement, hot weather also created additional concerns around health and safety for a few teachers

Page 25: The majority of participants discussed the impact of TDM on pupils’ learning, health and wellbeing. Conflict existed between participants’ perceptions of the impact on behaviour and concentration.

Page 27: Many pupils believed participating in TDM improved their attitude towards physical activity;

Page 28: Furthermore, some pupils and teachers attributed improvements in pupils’ sporting achievements to participating in TDM;

Page 28: Many pupils acknowledged the associations between physical activity and wellbeing;

Page 29: Many pupils reported a number of social benefits to participating in TDM. This included the opportunity to interact with peers and the positive subsequent effect during lessons/

There is very little discussion of the findings related to CRF in the context of other research e.g. are the authors able to comment on how the levels of fitness described in the current study compare to other studies on this population?

Response

Thank you for raising this point. I have now reframed the paper to focus the primary aim on the qualitative component. Therefore, the secondary aim examining CRF and discussions of findings have been shortened. In order to link the themes, I have discussed the findings in support of the qualitative reporting of perceived improvements in children’s CRF. In addition, I have addressed your query regarding discussing the CRF findings in relation to the literature. Please find details of this below. 

Change

Page 38: Teachers also noted perceived improvements in pupils’ CRF. This qualitative finding is supported by the exploratory analysis of the secondary aim conducted in this study which suggests that the CRF of children from South Wales increased between baseline and follow up following participation in TDM.

Page 39: In this sample of children from South Wales, UK, the deprived group performed a lower number of shuttles at both time points and thus, displayed a lower proportion of children classified as fit. The social gradient of physical activity, CRF and deprivation is demonstrated in the literature, with a larger proportion of children from a higher socio-economic status classified as fit compared to those from a lower socio-economic status [60]. Reducing inequalities in health and narrowing the deprivation gap in children’s CRF is a public health priority, given the wide range of health benefits of regular physical activity[1,3]. Indeed, there is widespread recognition that universal school-based programmes that engage children from a range of socioeconomic backgrounds are effective in improving pupil health and wellbeing [13].

There is a lot of repetition from the results section, consider making more concise. Are the authors able to tie in and align findings and discussion on CFR with implementation at all, at the moment it appears quite disjointed.

Response

Thank you for raising this point. Regarding your point of repetition, I have removed the first few paragraphs of the discussion where I believe the repetition existed. The purpose of these paragraphs were to provide a summary of findings in relation to implementation to the reader prior to discussing with the literature. However on reflection, I agree that these paragraphs are repetitive and the depth of the discussion remains with these removed. 

Since I have restructured the paper, I feel the discussion of CRF fits well as a secondary aim in support of the qualitative findings and participants’ view of improvements in CRF. 

Change

Page 33: Removed repetitive paragraphs from discussion (see tracked changes manuscript)

You have outlined a number of barriers and facilitators to implementation of TDM in school. It would be worthwhile also consulting with systematic reviews of schools based physical activity programs to expand this list and ensure it is consistent with recent literature. You may find the following references helpful when thinking about running programmes specifically:

Malden S, Doi L. The Daily Mile: teachers’ perspectives of the barriers and facilitators to the delivery of a school-based physical activity intervention. BMJ Open 2019;9:e027169. doi:10.1136/ bmjopen-2018-027169

Chalkley AE, Routen AC, Harris JP, Cale LA, Gorely T., Sherar LB. (2018) A retrospective qualitative evaluation of barriers and facilitators to the implementation of a school-based running programme. BMC Public Health, 18:1189

Response

Thank you for suggesting these papers and integrating their findings within this manuscript. Both papers have been a valuable addition to the literature featured in this manuscript and are incorporated within the Introduction and Discussion sections. See reference 30 for Chalkley et al. (2018) and 33 for Malden and Doi (2019). 

Change

Chalkley et al. (2018):

Page 5: Previous research into school-based running programmes has demonstrated the variability in implementation across schools[30].

Page 33-34: Headteachers, teachers and pupils discussed a range of factors associated with implementation in relation to the experience and engagement with TDM. These barriers and facilitators to effective implementation identified by participants are consistent with recent research into school-based running programmes[30,33]

Page 34: Furthermore, variation in implementation and flexibility has been documented as a facilitator in other school-based running programmes[30]. With flexibility consistently identified as a key factor to the effective implementation of interventions, it is essential for future programmes to be designed with this at the core.

Page 36: In this study, pupils currently not offered an afternoon playtime in school were positive about TDM providing a break from lessons. Research has highlighted the positive effect of active breaks on children’s cognitive function and academic achievement [30–32].

Page 39: In addition, previous research into school-based running programmes has identified unintended outcomes such as increased sedentary time outside of school compensating for the increased school PA[30]. However, no unintended outcomes were reported in this study. 

Malden and Doi (2019): 

Page 6: Two recent qualitative studies exploring the implementation processes and participants’ experiences of TDM identified a number of factors associated with intervention success[32,33]. These included a need for simple core intervention components, flexible delivery encouraging teacher autonomy and intervention adaptability. Benefits cited by teachers included improved teacher-pupil relationships and the positive impact on pupils’ health, wellbeing and fitness[33]. [33]. In contrast, a number of barriers were identified such as weather, resources and the perceived impact on learning time. Furthermore, the delivery style varied widely between school, warranting further investigation into how delivery affects participants’ experiences. These studies provide an important contribution to the understanding of implementation and experiences of TDM

Page 10: In order to explore participants’ experiences of TDM, it is important to consider the barriers, facilitators and factors affecting sustainability. These factors are consistently included in other research evaluating school-based interventions, and therefore framed the topics guides for this study [33]. 

Page 33: However, limited research exists examining the implementation and experience of TDM from a whole-school perspective [32,33]

Page 34: Previous qualitative research exploring implementation of TDM highlighted the importance of flexible implementation in facilitating teacher autonomy and engagement [32,33]

Page 35: The most significant barrier to implementation identified by participants was that of curriculum pressure, as cited in recent research on TDM [33].

Page 35: However, the suggestion that Daily Mile could possibly be being used as a replacement to PE is concerning and has been cited in another qualitative study of TDM [33].

Page 36: However, recent research into the implementation of TDM[32] suggests that this caused less disruption to the school day and given the impact on learning time cited in this and other studies[33], headteachers may feel they have no choice.

Page 37: Furthermore, teacher participation may elicit wider benefits such as improved teacher-pupil rapport, as identified in the literature[33].

Page 38: Overall, views on the effect of TDM on pupils’ behaviour and concentration were mixed. Some pupils felt their ability to concentrate in lessons and attitude to learning improved following participation in TDM. These immediate effects are supported by other qualitative research exploring implementation of TDM[32,33]

Are the authors able to suggest ways future research can confirm findings and or build on what has been learnt? The authors state that the current study is a pilot study (line 29), which implies that the results will inform a full-trial. I would question what scope there is for a full trial of the intervention if it has already been implemented. It could be that it is being phased out region by region, in which case it would be possible to do a pilot and then a larger scale evaluation in a new area, but it should be clear in the paper

Response

We appreciate this query and acknowledge that a full trial would not be feasible given the demands associated with establishing implementation levels across schools in addition to the widespread uptake. Given the restructure of the manuscript focussing on the qualitative findings and the addition of the change below, the authors feel that this has been addressed. 

Change

Page 39-40: Future research would benefit from an in-depth process evaluation of a larger sample of schools. Triangulation of findings could help highlight of the strength and weaknesses of implementation factors on outcomes.

A number of practical recommendations to support the implementation of running programmes has been provided and the authors may find it useful to refer to these in the discussion

Response

Thank you for suggesting the ‘how to guide’. Please find details of inclusion of this guide below (reference 25).

Change

Page 5: Furthermore, a ‘how to guide’ has been published by the University of Stirling as an outline for schools regarding implementation and research findings[25].

Page 37: The inclusion of teachers has also been advocated in a ‘how to guide’ developed by the University of Stirling, in which teacher participation and informal communication with pupils is encouraged [25].

Page 37: In addition, although one of the benefits cited by TDM is the lack of clothing or equipment required [25], this posed a challenge to schools.

The natural experiment design has the potential to introduce bias. Some acknowledgements of this bias as a limitation would be worthwhile as well as the design’s ability to isolate the effect on CRF and the numerous other variables which could influence it e.g. maturation, seasonality, physical activity levels etc

Response

Thank you for raising this point. A natural experimental approach was viewed as the most suitable approach given the widespread uptake of TDM and this approach is often utilised in school-based research. However as you have quite rightly stated, this design poses a number of limitations. Please see below acknowledgement of these. 

Change

Page 40: School-based research poses a number of challenges in relation to the recruitment of schools. In this research study, a natural experimental approach was chosen due to the widespread adopted of TDM as a result of political and media support. However, the lack of a control group creates challenges in concluding the direct effect of TDM on children’s CRF. Recruitment for this research study was conducted through convenience sampling, in which schools chose to begin implementing TDM at different time-points to coincide with school terms. This convenience sampling method could elicit selection bias as schools that volunteered to participate in the research study are likely to have a greater interest and investment in TDM with the potential to generate more positive feedback on implementation. 

Page 40: In this research study, data collection was completed in two data collection phases due to schools choosing to implement TDM at different time-points. Research has identified the effect of seasons on PA, with lower levels of MVPA exhibited in autumn and winter [61]. Statistical analyses did not adjust for season and this should be taken into account when interpreting findings.

Page 40: Finally, it must be considered that changes in CRF could be due to a number of other factors aside from TDM such as growth and maturation [62] and improvement in 20m SRT participation.

Convenience sampling schools is likely to introduce some bias. Acknowledgement of this and what would be deemed as best practice is required.

Response

Thank you for suggesting that I acknowledge this limitation. Convenience sampling was the most suitable sampling strategy to employ in this research study. However, the authors acknowledge it is likely to introduce bias and this has been addressed in the Strengths and Limitations section. 

Change

Page 40: Recruitment for this research study was conducted through convenience sampling, in which schools chose to begin implementing TDM at different time-points to coincide with school terms. This convenience sampling method could elicit selection bias as schools that volunteered to participate in the research study are likely to have a greater interest and investment in TDM with the potential to generate more positive feedback on implementation. 

In addition, if I have understood it right, the differing length of exposure to TDM should be acknowledged i.e. School A was implementing for 6 months, schools C-F for 5 months and School B for 2 months, and implications of this on the quantitative and qualitative data collected e.g. limits ability to establish how such programmes are maintained in schools.

Response

Thank you for raising this. Due to the natural experiment approach and convenience sampling method employed, schools chose when to begin implementing TDM to align with school terms (School A – start of spring term, School B – start of summer term, School C-F – start of autumn term). I have provided clarity on this for the reader, in addition to the other changes below: 

Change

Page 7-8: A natural experimental approach is considered the most suitable methodology when intervention implementation cannot be controlled by the researcher [34]. In the case of this research study, this was due to the rapid adoption of TDM encouraged by media and political attention[35]. In this research study, schools began delivering TDM at three time-points aligned with academic terms (School A – January 2017, start of spring term, School B – May 2017, start of summer term, School C-F – September/October 2017, start of autumn term). Data collection was completed in two phases to reflect the two academic years (Phase one 2016-17- School A and B, Phase two 2017-18 – School C-F). Data collection was conducted at two time-points; baseline (before implementation) and follow-up (3-6 months post implementation). A schematic diagram representing data collection periods across schools A-F is provided for clarity in the supporting information (S1 Appendix).

Page 34: Conflict existed between schools on how TDM should be delivered, raising the issue of fidelity to the intervention.

Page 34: However from a research perspective, variation in delivery and fidelity to the original intervention design poses a number of challenges for evaluating school-based interventions.

Page 40: The implementation and fidelity of TDM was not directed measured in this study, although anecdotal differences are reported through qualitative findings. 

I do not feel as if the results are supportive of the conclusion in relation to CRF due to the reasons outlined previously.

Response

Thank you for raising your concerns regarding the conclusions of CRF. The authors agree that given the limitations to the study design, the conclusions on CRF need to be weakened. To address these, I have removed strong conclusions from the manuscript and acknowledged the limitations for the reader to be aware of when drawing their conclusions. 

Change

Page 39: As implementation and adherence was not directly measured, in addition to limitations with the study design, it is difficult to draw conclusions on the overall effect of TDM on children’s CRF.

Please check your referencing system. There are missing references e.g. no 40 (line 724) as well as inappropriate references provided.

Response

Thank you for drawing attention to issues with the referencing system. I can confirm that authors have proofread the referencing and amended any mistakes. 

Change

Page 42: Updated referencing system

I would suggest reordering the guides in the supplementary files so that baseline interview/focus group guide is presented first.

Response

Thank you for this suggestion. I agree that reordering the supplementary files in order of time point would provide clarity to the reader.

Change

Updated supporting information (S3 Appendix)

Reviewer #2

I would like to suggest the authors consider flipping the focus of the manuscript to the qualitative aspect, with far less emphasis placed on the CRF data presented. My reasons for this suggestion and other specific comments can be found below.

Response

Thank you for this valued suggestion. I agree that the manuscript would be strengthened by changing the focus to the qualitative findings. As such, I have restructured the manuscript to focus on the primary aim; exploring participants’ experiences of The Daily Mile (TDM) to gain an understanding of how implementation is related to experience. I have included the quantitative element as a secondary. In addition, I have linked the secondary aim in relation to supporting the qualitative findings of perceived improvements in children’s CRF. 

Please find restructuring throughout the manuscript. 

Change

Page 6-7: The primary aim of this mixed-methods study was to explore the pupils’, teachers’ and headteachers’ experiences of The Daily Mile and understand whether experience was related to implementation. The secondary aim of this study was to examine the association between The Daily Mile and children’s cardiorespiratory fitness and compare this association between children in high and low socio-economic groups. 

“Both groups demonstrated equal increases in shuttles between baseline and follow-up (deprived: 4.7 ± 13.4, nondeprived: 4.8 ± 16.0). There was no significant difference in the increase in shuttles run for deprived compared to non-deprived children adjusted for age and gender. This finding suggests that TDM is an effective universal tool in improving children’s CRF.” Given the relatively small change in 20mSRT performance (less than half a level on the shuttle run test) and the large SD associated with the change in both groups (highlighted above), I do not think the statement the ‘TDM is an effective universal tool in improving children’s CRF’ is supported. I suggest softening greatly, and also acknowledging the sample (children from South Wales) within any statement

Response

Thank you for raising your concerns with this conclusion. I agree that this was strongly worded given the large standard deviation. To address this, I have removed this from the manuscript and I have softened all conclusions on the quantitative findings. Any discussions of the quantitative data are now referred to in relation to the sample of children from South Wales within the discussion on the quantitative findings.

Change

Page 30: Both groups demonstrated equal increases in shuttles between baseline and follow-up (deprived: 4.7 ± 13.4, non-deprived: 4.8 ± 16.0). However, these results exhibit large standard deviation.

Page 38: In this sample of children from South Wales, UK, the deprived group performed a lower number of shuttles at both time points and thus, displayed a lower proportion of children classified as fit.

If the purpose of the introduction is to provide rationale for the need to increase physical activity levels, then the first few paragraphs of the introduction are very strong. The focus of the paper however seems to be on cardiorespiratory fitness, which (as I’m sure the authors know) is distinctly different from physical activity. The fact that cardiorespiratory fitness is not mentioned until the aims of the study are listed concerns me, as no rationale for examining this outcome has really been provided. The introduction reads as though it is physical activity that is going to be measured, not cardiorespiratory fitness. This should be addressed in the revision process and I suggest the authors consider directing the focus of this manuscript away from the CRF element.

Response

Thank you for raising this valid point. The design of TDM is a children’s running programme to increase physical activity and fitness. However, I appreciate that the Introduction within this manuscript discusses solely physical activity, with no reference to children’s cardiorespiratory fitness (CRF) until the aims of the study are listed. I have addressed this throughout the Introduction section by referring to the point of PA acting as the principle, modifiable determinant of CRF. Please find further details of these changes below.

Changes

Page 2: PA is also the principle, modifiable determinant of cardiorespiratory fitness (CRF)[4], which reflects the cardiovascular and respiratory system’s capacity to supply oxygen during long-term PA[5]. Higher levels of CRF during childhood have been associated with a range of positive health outcomes similar to those of regular PA such as cardiovascular health. Research has demonstrated the relationship between PA and CRF in children regardless of gender, age, ethnicity, economic status and school[4]. 

Page 4: However, accurately measuring children’s PA levels presents a number of methodological limitations[10]. Self-report methods including questionnaires are associated with subjectivity issues such as recall bias and are not advised in children younger than 10 due to their limited ability to accurately report PA[10]. On the other hand, whilst objective measurements such as accelerometry can measure PA across the domains of frequency, intensity and duration, they require participant adherence and are high-cost for researchers. Thus, as increasing levels of PA in childhood improves CRF and higher levels of CRF are associated with positive health outcomes[11], measuring CRF in children through methods such as the 20m shuttle run test (20m SRT) provides a valid, low-cost and pragmatic approach to assessing health-related PA interventions[12].

Page 4: This expansion was partly encouraged by rapid media and government attention, despite at the time no published evidence existing regarding its anecdotal benefits such as improved CRF, behaviour and concentration.

The paper cited in reference 13 (e.g. Chesham RA, Booth JN, Sweeney EL, Ryde GC, Gorely T. The Daily Mile makes primary school children more active , less sedentary and improves their fitness and body composition : a quasi-experimental pilot study. BMC Med. 2018) has been heavily critiqued and the findings discredited by several sources due to major limitations in the study design. Please consider and discuss these critiques in a lot more detail for transparency.

Response

Thank you for raising the concerns with this paper. 

Change

Page 5: However, this study has been widely critiqued due to methodological weaknesses such as a small sample size. In a response, Daly-Smith et al [24] suggest a more cautious interpretation of these conclusions is required and call for further evidence of TDM in establishing an understanding of its impact, both positive and negative.

Can the authors please provide some more context as to why it is important to look at deprived and non-deprived cohorts separately? Please contextualise this to Wales in particular?

Response

Thank you for this point. TDM is a universal school-based running intervention that has achieved widespread adoption despite limited research examining its impact. To date, no research has looked at its impact on different socioeconomic groups. There is potential for universal interventions to widen inequalities (the inequality paradox). Given the wide socioeconomic spread of schools within the area of this research study, the authors felt that this would be a useful addition to the literature. In addition, the authors felt that a breakdown of the associations of TDM on children’s CRF by socioeconomic status to understand if the effects of TDM are universal across all groups would be welcomed in the literature to understand whether TDM is universally beneficial or does it favour one socioeconomic group (e.g. affluent, more active children). However, I have included an ‘overall’ column within the quantitative results section to provide a further clarity on results for the reader. 

Change

Page 32: Table 2 updated with overall column 

Please strongly consider changing the focus of the manuscript to the second aim (e.g. explore whether children’s experiences of The Daily Mile was related to implementation) and mention CRF as an exploratory secondary outcome only. The qualitative element is so strong and I feel the CRF element detracts from this excellent work and weakens the paper overall.

Response

Thank you for your compliments regarding the qualitative aspect of this manuscript. The authors agree that this component is significantly stronger and reframing the paper to focus on this as the primary outcome would really strengthen the paper and its contribution to the literature. I have restructured the paper and changed the primary aim to explore pupils’, teachers’ and headteachers’ experiences of TDM and understand how experience was related to implementation. I have adjusted the ordering of sections, in particular the methodology section to reflect this. Furthermore, I have focussed the majority of the discussion around the qualitative findings in relation to the literature. The quantitative component is now a secondary aim. Findings from this are discussed with less focus and results supporting the qualitative findings of perceived improvements in CRF.

Personally, I do not think the study design utilised here allows for such strong conclusions on CRF to be made. This should be recognised in study design section and is the main reason I do not think CRF should be the focus of this manuscript. 

Response

Thank you for raising your concerns regarding the study design. I agree that the natural experimental approach, convenience sampling method and different data collection time points generate challenges in concluding the effects of TDM on children’s CRF. To address this concern, I have made the qualitative component the primary aim of this research study. I feel this significantly contributes to the literature regarding implementation of TDM. I have restructured for methodology to reflect this. Furthermore, I have weakened statements regarding CRF and acknowledged the methodological weaknesses within this study in the ‘Strengths and limitations’ section. 

Change

Page 38-39: This qualitative finding is supported by the exploratory analysis of the secondary aim conducted in this study which suggests that the CRF of children from South Wales increased between baseline and follow up following participation in TDM. An equal increase in the number of shuttles run by the deprived and non-deprived groups was observed and adjusting for age and gender, there was no significant difference between children of high and low socio-economic groups. However, these results exhibit large standard deviation and therefore, strong conclusions on the association between TDM and children’s CRF cannot be made.

Page 39: As implementation and adherence was not directly measured, in addition to limitations with the study design, it is difficult to draw conclusions on the overall effect of TDM on children’s CRF.

While conducting a natural experiment is pragmatic and should be commended, limitations within the study design does not allow confidence in the conclusions drawn from this study. Firstly, as of course there is no control group, it is impossible to ascertain whether the pupils involved in study increased their CRF due to TDM, or because of some other factors (e.g. growth and maturation, measurement error within the test, seasonal effects, or pure coincidence). This has to be addressed throughout the manuscript and conclusions adapted throughout

Response

Thank you for discussing your concerns regarding the quantitative component of this research study. As you have mentioned, a natural experimental approach in school-based research is often the most pragmatic approach to employ. However, this approach brings with it a number of limitations which I have acknowledged below. 

As I have restructured the paper to focus on the primary qualitative aim, the quantitative results and discussion of CRF have been shortened and addressed in relation to the qualitative findings. 

Change

Page 39-40: School-based research poses a number of challenges in relation to the recruitment of schools. In this research study, a natural experimental approach was chosen due to the widespread adopted of TDM as a result of political and media support. However, the lack of a control group creates challenges in concluding the direct effect of TDM on children’s CRF. Recruitment for this research study was conducted through convenience sampling, in which schools chose to begin implementing TDM at different time-points to coincide with school terms. This convenience sampling method could elicit selection bias as schools that volunteered to participate in the research study are likely to have a greater interest and investment in TDM with the potential to generate more positive feedback on implementation. In this research study, data collection was completed in two data collection phases due to schools choosing to implement TDM at different time-points. Research has identified the effect of seasons on PA, with lower levels of MVPA exhibited in autumn and winter[62]. Statistical analyses did not adjust for season and this should be taken into account when interpreting findings. However, the strength of this from a qualitative perspective is that experiences are captured throughout the academic year. Furthermore, schools had varying exposure to TDM and a dose-response relationship may impact on CRF. Finally, it must be considered that changes in CRF could be due to a number of other factors aside from TDM such as growth and maturation[63] and improvement in 20m SRT participation.

I also think it should be made clearer why post-intervention measures were collected at three months in some schools and six months in others. In terms of dose, this suggests some schools had double the dose of the intervention than others, which likely will have impacted the findings. Again, I am aware the pragmatic nature of the study design may have dictated this, but this is another reason for weakening the statements on CRF improvement and changing the manuscript focus towards implementation instead.

Response

Thank you for raising this point regarding exposure to TDM. Due to the natural experiment approach and convenience sampling method employed, schools chose when to begin implementing TDM to align with school terms (School A – start of spring term, School B – start of summer term, School C-F – start of autumn term). Data collection was conducted to accommodate this. I have provided clarity on this for the reader, in addition to the other changes below: 

Change

Page 7-8: A natural experimental approach is considered the most suitable methodology when intervention implementation cannot be controlled by the researcher [34]. In the case of this research study, this was due to the rapid adoption of TDM encouraged by media and political attention[35]. In this research study, schools began delivering TDM at three time-points aligned with academic terms (School A – January 2017, start of spring term, School B – May 2017, start of summer term, School C-F – September/October 2017, start of autumn term). Data collection was completed in two phases to reflect the two academic years (Phase one 2016-17- School A and B, Phase two 2017-18 – School C-F). Data collection was conducted at two time-points; baseline (before implementation) and follow-up (3-6 months post implementation). A schematic diagram representing data collection periods across schools A-F is provided for clarity in the supporting information (S1 Appendix).

Page 40: In this research study, data collection was completed in two data collection phases due to schools choosing to implement TDM at different time-points. Research has identified the effect of seasons on PA, with lower levels of MVPA exhibited in autumn and winter[62]. Statistical analyses did not adjust for season and this should be taken into account when interpreting findings. However, the strength of this from a qualitative perspective is that experiences are captured throughout the academic year. Furthermore, schools had varying exposure to TDM and a dose-response relationship may impact on CRF. 

Can you please provide information on how pupils were recruited into the focus group part of the study? Was consent for this element separate from the 20mSRT?

Response

Thank you for raising this point about focus group recruitment and consent. Pupils had the opportunity to consent individually to participation in focus groups and the 20m SRT within one consent form

Change

Page 9: All pupils from years 5&6 from schools A-F were invited to participate in both the qualitative and quantitative measures. Pupils had the option to consent to participate in one or both measures in consent forms. Headteachers and all teachers from years 5&6 from the six schools were invited to participate in the qualitative measure

Page 10: A further breakdown of interviews and focus group participation by school can be found in the supporting information (S2 Appendix). 

Can you also indicate whether all pupils involved in the 20mSRT had the opportunity to take part in the focus groups, or whether a sub-sample were recruited? If the latter, how was this achieved?

Response

Thank you for requesting clarity on this point. As detailed above, all pupils were invited to take part in the 20m SRT and focus group. Consent was provided individually for each component on consent forms. Focus group selection was carried out as outlined below.

Change

Page 10: Each focus group was conducted by year group and consisted of between six and eight pupils [42] aged 9-11 years of mixed physical activity ability and gender. Class teachers were provided with a list of consented pupils and selected pupils fulfilling this criteria. This list was discarded following selection of pupils and a final list of pupils participating in focus groups was not recorded to ensure anonymity

Please provide more detail in the steps conducted for the qualitative synthesis of the data.

Response

Thank you for bringing this to my attention. I agree that further clarity is needed on the qualitative analysis process. I have further described the steps that were taken in this research study below, outlined by Burnard (1991):

Change

Page 12: The process of analysing the interview and focus group data followed the steps outlined by Burnard (1991)[48]. To begin, each transcript was independently read several times by two researchers (EM and CT) to facilitate immersion in the data. The researchers (EM and CT) then followed an independent open coding process to allow participants’ views to be summarized by assigning words or phrases to quotes or paragraphs. This initial list of freely generated categories following review of the transcripts aimed to encapsulate interviewees’ responses and were subsequently grouped according to the overarching theme. Through this process, broader categories were combined to produce one higher-order heading that captured the overall meaning of responses. This process was repeated whereby similar categories were synthesised to produce a final list of themes and sub-themes. Both researchers (EM and CT) compared their lists of themes and sub-themes to ensure accuracy and consistency. If there was a discrepancy or disagreement in coding, a third researcher (SB) adjudicated. This method enhances the validity of categories assigned and attempts to reduce researcher bias [48]. The written notes taken on the day of the interview or focus group were compared with these topics to ensure an accurate account of participants’ responses. Following this, the two researchers worked together through an extensive process to discuss codes and categorise them under final themes and sub-themes (S4 Appendix). The lead researcher (EM) then manually worked through each transcript and coded the responses according to the final list of themes and sub-themes. All responses grouped by themes and sub-themes were compiled to a master copy document that was used for reference to write up the findings. 

“However, the number of children who were classed as fit in the deprived group increased by 14% compared to an 8% increase in the non-deprived children”. I am not sure how relevant this statement is, as surely it is obvious more children in the deprived group were classed as fit post-intervention, due to the differences between the groups at baseline? E.g. more kids in the non-deprived group were already in the ‘fit’ group as baseline, therefore could not progress to a ‘higher’ group as there was nowhere to go?

Response

This is a very interesting point and thank you for raising it. The results demonstrate the difference in fit classification between deprived and non-deprived groups, with a lower proportion of children classified as fit in the non-deprived group at both time points. However, given that there was a lower proportion of children in the non-deprived group at baseline, I agree with your point that there was a greater likelihood in their ability to move up a group (to fit classification).

Change

To address your concerns, I have removed this statement from the manuscript. Instead, I have described the differences in fit classification by deprived group in relation to the literature. 

Page 39: In this sample of children from South Wales, UK, the deprived group performed a lower number of shuttles at both time points and thus, displayed a lower proportion of children classified as fit. The social gradient of physical activity, CRF and deprivation is demonstrated in the literature, with a larger proportion of children from a higher socio-economic status classified as fit compared to those from a lower socio-economic status [60]. Reducing inequalities in health and narrowing the deprivation gap in children’s CRF is a public health priority, given the wide range of health benefits of regular physical activity[1,3].

I would however like the authors to comment on the whether a change in ~5 shuttles over a 3-6 month programme is actually meaningful. I would also like the authors to comment on the large SD associated with the change in CRF, and the impact this has on the interpretation of the findings.

Response

Thank you for raising this query regarding the change in shuttles from baseline to follow up. A change in ~5 shuttles would equate to an additional 100m run in the 20m SRT. Given the increase from 49% of children classified as fit at baseline overall to 60%, this increase was meaningful enough to increase the number of children classified as fit. However as you have pointed out, the large standard deviation presents caution in interpreting findings and drawing strong conclusions. As a result, I have acknowledged the large standard deviation and have addressed the limitations in concluding the effect of TDM on children’s CRF. 

Change

Page 30: Both groups demonstrated equal increases in shuttles between baseline and follow-up (deprived: 4.7 ± 13.4, non-deprived: 4.8 ± 16.0). However, these results exhibit large standard deviation.

Page 39: However, these results exhibit large standard deviation and therefore, strong conclusions on the association between TDM and children’s CRF cannot be made.

---

## [Decision Letter · Decision Letter 1]

13 Dec 2019

PONE-D-19-21022R1

The Daily Mile: whole-school recommendations for implementation and sustainability. A mixed-methods study

PLOS ONE

Dear Miss Marchant,

Thank you for submitting your manuscript to PLOS ONE. After careful consideration, we feel that it has merit but does not fully meet PLOS ONE’s publication criteria as it currently stands. Therefore, we invite you to submit a revised version of the manuscript that addresses the points raised during the review process.

ACADEMIC EDITOR: Thank you for taking the time and effort to address the previous comments from the reviewers. We are all agreed that the manuscript is greatly improved as a result. The reviewers have some relatively minor issues that they would like to see addressed before the manuscript is ready for publication; reviewer 1 has identified some typographical errors, while reviewer 2 has asked if you could add confidence intervals and provide further detail on how you sampled pupils into the focus groups. These should be very quick and easy for you to address so I'll look forward to seeing the revised version soon.

We would appreciate receiving your revised manuscript by Jan 27 2020 11:59PM. To enhance the reproducibility of your results, we recommend that if applicable you deposit your laboratory protocols in protocols.io, where a protocol can be assigned its own identifier (DOI) such that it can be cited independently in the future. For instructions see: http://journals.plos.org/plosone/s/submission-guidelines#loc-laboratory-protocols

We look forward to receiving your revised manuscript.

Kind regards,

Shelina Visram, PhD, MPH, BA

Academic Editor

PLOS ONE

Reviewers' comments:

Reviewer's Responses to Questions

**Comments to the Author**

1. If the authors have adequately addressed your comments raised in a previous round of review and you feel that this manuscript is now acceptable for publication, you may indicate that here to bypass the “Comments to the Author” section, enter your conflict of interest statement in the “Confidential to Editor” section, and submit your "Accept" recommendation.

Reviewer #1: All comments have been addressed

Reviewer #2: (No Response)

2. Is the manuscript technically sound, and do the data support the conclusions?

Reviewer #1: Yes

Reviewer #2: Partly

3. Has the statistical analysis been performed appropriately and rigorously? 

Reviewer #1: Yes

Reviewer #2: Yes

4. Have the authors made all data underlying the findings in their manuscript fully available?

Reviewer #1: Yes

Reviewer #2: Yes

5. Is the manuscript presented in an intelligible fashion and written in standard English?

Reviewer #1: Yes

Reviewer #2: Yes

6. Review Comments to the Author

Reviewer #1: Thank you for your comprehensive response to the reviewers comments. I am satisfied you have addressed my comments, and I feel the manuscript is improved as a result. There are just a few minor amends to address.

Introduction - line 65 - 'the' missing from the sentence before 'latest' ?

Quantitative results p 31 line 744 - please could you insert the percentages for number of pupils in brackets

Discussion - line 785-787 - Please review this sentence, currently it doesn't make sense

Strengths and limitations - line 1005 amend 'adopted' to 'adoption'

Reviewer #2: I would firstly like to congratulate the authors on their responses and revision of this paper. The manuscript is much improved and the authors should be commended on their drive to provide transparent and rigorous science on such a politically charged intervention. I have two remaining queries that I would like the authors to address. The first is with regards to the presentation of the CRF data. I am glad the authors have now refocused their work away from this outcome. For further transparency, I would like the authors to consider adding 95% confidence intervals to their CRF changes, as this will further highlight the wide variability in the change in CRF scores and allows the reader to easily see this. Once this change has been made in the manuscript, I also request that this information is included in the abstract and if word count allows, a few words added to highlight the variability in the data and the implications of this.

My second question relates to how the focus group participants were chosen. Am I correct in thinking the class teachers chose the school pupils based on the list of those who provided consent? If so, could this be viewed as 'cherry picking', in that the teachers may have chosen pupils based on their knowledge of them (e.g. choosing pupils that they thought enjoyed the programme/ were likely to talk positively about it/ were already 'good standing' pupils within the school community). I would like the authors to consider this point and add this potential bias to the limitations of the study.

7. PLOS authors have the option to publish the peer review history of their article (what does this mean?). If published, this will include your full peer review and any attached files.

Reviewer #1: No

Reviewer #2: Yes: Kathryn Weston

---

## [Author Response · Author response to Decision Letter 1]

6 Jan 2020

Reviewer #1

Thank you for your comprehensive response to the reviewers comments. I am satisfied you have addressed my comments, and I feel the manuscript is improved as a result. There are just a few minor amends to address.

Introduction - line 65 - 'the' missing from the sentence before 'latest'?

Response

Thank you for bringing this oversight to my attention. I have addressed this as outlined below. 

Change

Page 3: Furthermore, survey level data from the latest Active Healthy Kids Wales Report Card within Wales, United Kingdom suggests that just 34% of children aged 3-17 years are meeting these guidelines[9] 

Quantitative results p 31 line 744 - please could you insert the percentages for number of pupils in brackets

Response

I agree that displaying the percentage of eligible pupils that participated in the measures would provide further clarity to the reader. I have updated this within the manuscript. 

Change

Page 30: There was a total of 336 pupils in years 5 and 6 attending the six primary schools in this study. From this sample of eligible pupils, 229 pupils (68%) participated in the 20m SRT at baseline and 235 pupils (70%) at follow up. In total, 204 pupils (61%) completed the 20m SRT at both time points.

Discussion - line 785-787 - Please review this sentence, currently it doesn't make sense

Response

Thank you for suggesting a restructure of this sentence, I agree that it was not clear I have addressed your concerns with the following change:

Change

Page 34: Removed original sentence

Page 34: After the initial excitement of a new school activity, pupils commented on feeling bored and lacking enjoyment, impacting on pupils’ participation and the longer-term sustainability of TDM

Strengths and limitations - line 1005 amend 'adopted' to 'adoption'

Response

Thank you again for raising this oversight. I have made the amendment as instructed.

Change

Page 40: In this research study, a natural experimental approach was chosen due to the widespread adoption of TDM as a result of political and media support. 

Reviewer #2

I would firstly like to congratulate the authors on their responses and revision of this paper. The manuscript is much improved and the authors should be commended on their drive to provide transparent and rigorous science on such a politically charged intervention. I have two remaining queries that I would like the authors to address.

Response

The first is with regards to the presentation of the CRF data. I am glad the authors have now refocused their work away from this outcome. For further transparency, I would like the authors to consider adding 95% confidence intervals to their CRF changes, as this will further highlight the wide variability in the change in CRF scores and allows the reader to easily see this. Once this change has been made in the manuscript, I also request that this information is included in the abstract and if word count allows, a few words added to highlight the variability in the data and the implications of this. 

Response

Thank you for your suggestion of adding in 95% confidence intervals to the CRF changes data. I agree that this would provide clarity on the variability within the data. I have included the 95% confidence intervals within the manuscript and have highlighted the wide variability within the results and discussion section. As the abstract is already at the word limit, I am unable to include this information in the abstract. 

Change

Page 31: However, these results exhibit large standard deviation and wide 95% confidence intervals (deprived: 2.0 to 7.4, non-deprived: 2.3 to 7.3), demonstrating the variability that is present among this sample

Page 32: Table 2 updated with confidence intervals 

Page 38-39: An equal increase in the number of shuttles run by the deprived and non-deprived groups was observed, however, the wide confidence intervals present within this data demonstrate the variability of changes in children’s CRF. 

My second question relates to how the focus group participants were chosen. Am I correct in thinking the class teachers chose the school pupils based on the list of those who provided consent? If so, could this be viewed as 'cherry picking', in that the teachers may have chosen pupils based on their knowledge of them (e.g. choosing pupils that they thought enjoyed the programme/ were likely to talk positively about it/ were already 'good standing' pupils within the school community). I would like the authors to consider this point and add this potential bias to the limitations of the study

Response

Thank you for requesting further information on the sampling for focus groups within this study. I agree that this information is important for the reader. I understand your concerns regarding the potential for ‘cherry picking’, however, teachers were reminded of the importance of selecting children that fulfilled the criteria (e.g. mixed physical activity ability). Having worked closely with the schools and teachers within this study, the authors have confidence that this was achieved. To address your queries, I have explained this further in the results section and in the strengths and limitations section as outlined below.

Change

Page 10: Teachers were reminded of the need to include pupils of a range of physical activity abilities.

Page 41: Focus group sampling was achieved through teachers selecting consented pupils fulfilling a criteria of mixed gender and physical activity abilities. Although teachers were reminded of the importance of selecting a variety of pupils with a range of abilities in order to capture a variety of experiences, there is potential that this method could cause bias in results through preferential selection.

---

## [Editor Report · Decision Letter 2]

9 Jan 2020

The Daily Mile: whole-school recommendations for implementation and sustainability. A mixed-methods study

PONE-D-19-21022R2

Dear Dr. Marchant,

We are pleased to inform you that your manuscript has been judged scientifically suitable for publication and will be formally accepted for publication once it complies with all outstanding technical requirements.

With kind regards,

Shelina Visram, PhD, MPH, BA

Academic Editor

PLOS ONE
---

## [Editor Report · Acceptance letter]

14 Jan 2020

PONE-D-19-21022R2 

The Daily Mile: whole-school recommendations for implementation and sustainability. A mixed-methods study. 

Dear Dr. Marchant:

I am pleased to inform you that your manuscript has been deemed suitable for publication in PLOS ONE. Congratulations! Your manuscript is now with our production department. 

With kind regards,

on behalf of

Dr. Shelina Visram 

Academic Editor

PLOS ONE